# Strong charge-photon coupling in planar germanium enabled by granular aluminium superinductors

Marián Janík [1,2] ✉, Kevin Roux [1], Carla Borja-Espinosa[1], Oliver Sagi[1], Abdulhamid Baghdadi[1], Thomas Adletzberger[1], Stefano Calcaterra [3], Marc Botifoll [4], Alba Garzón Manjón[4], Jordi Arbiol [4,5], Daniel Chrastina [3], Giovanni Isella [3], Ioan M. Pop [6,7,8] & Georgios Katsaros [1]

High kinetic inductance superconductors are gaining increasing interest for the realisation of qubits, amplifiers and detectors. Moreover, thanks to their high impedance, quantum buses made of such materials enable large zero-point fluctuations of the voltage, boosting the coupling rates to spin and charge qubits. However, fully exploiting the potential of disordered or granular superconductors is challenging, as their inductance and, therefore, impedance at high values are difficult to control. Here, we report a reproducible fabrication of granular aluminium resonators by developing a wireless ohmmeter, which allows in situ measurements during film deposition and, therefore, control of the kinetic inductance of granular aluminium films. Reproducible fabrication of circuits with impedances (inductances) exceeding 13 k$\Omega$ (1 nH per square) is now possible. By integrating a 7.9 k$\Omega$ resonator with a germanium double quantum dot, we demonstrate strong charge-photon coupling with a rate of $g_c/2\pi$ = 566 ± 2 MHz. This broadly applicable method opens the path for novel qubits and high-fidelity, long-distance two-qubit gates.

Superconducting circuits made from disordered superconductors with large kinetic inductance are attracting attention for the development of qubits, such as fluxonium[1–5], parametric amplifiers[6–8] and kinetic inductance detectors[9–14]. High characteristic impedance devices, enabled by large kinetic inductors, are also paramount to semiconductor quantum dot qubit applications, whereby several studies have demonstrated strong charge-photon[15–18] and spin-photon coupling[19–22]. Furthermore, the first long-distance coherent coupling experiments between spin qubits have been reported[23,24], leading to the first photon-mediated long-distance two-qubit gate for electron spins in Si[25]. Currently, one of the main challenges for improving the

fidelity of such gates lies in increasing the spin-photon coupling strength[25]. This can be achieved by enhancing the charge-photon coupling and the spin-orbit interaction.

The charge-photon coupling strength $g_c$ is enhanced with high-impedance $Z$ resonators, as $g_c \propto \sqrt{Z}$[3,26–30]. To date, quantum dot circuit quantum electrodynamics (cQED) implementations utilising disordered nitride[21,22,25,31] or Josephson junction array[17,18] resonators have not exceeded the characteristic impedance of ∼ 3.8 k$\Omega$ (Supplementary Fig. 9). Outside the field of quantum dot cQED devices, much higher impedance has been achieved, either relying on the combination of a large kinetic inductance with a low stray capacitance of a

[1]ISTA, Institute of Science and Technology Austria, Am Campus 1, 3400 Klosterneuburg, Austria. [2]Institute of Electrical Engineering, Slovak Academy of Sciences, 841 04 Bratislava, Slovakia. [3]L-NESS, Physics Department, Politecnico di Milano, via Anzani 42, 22100 Como, Italy. [4]Catalan Institute of Nanoscience and Nanotechnology (ICN2), CSIC and BIST, Campus UAB, Bellaterra, 08193 Barcelona, Catalonia, Spain. [5]ICREA, Passeig de Lluís Companys 23, 08010 Barcelona, Catalonia, Spain. [6]IQMT, Karlsruhe Institute of Technology, 76131 Karslruhe, Germany. [7]PHI, Karlsruhe Institute of Technology, 76131 Karlsruhe, Germany. [8]Physics Institute 1, Stuttgart University, 70569 Stuttgart, Germany. ✉e-mail: marian.janik@ista.ac.at

suspended Josephson junction array with an impedance of $> 200\,k\Omega$[32], or a large mutual geometric inductance with a low stray capacitance of a suspended planar coil with ~ 31 kΩ[33]. These approaches cannot be readily implemented with standard semiconductor spin qubits devices because of incompatibility with magnetic fields and complex nano-fabrication requirements. Other studies have focused on the combination of a kinetic and mutual geometric inductance of meandered structures of thin-film TiN, with a kinetic inductance of 234 pH per square ($\square^{-1}$)[34], or granular aluminium (grAl) with 220 pH $\square^{-1}$[13], both with $Z \approx 28\,k\Omega$. While grAl offers magnetic field resilience[35], lift-off compatibility[1], low losses[36], and sheet kinetic inductance reaching 2 nH $\square^{-1}$[37–39], it was not yet exploited for quantum dot devices. This is presumably because of the poor reproducibility of high-impedance films[40], as the film resistance, which determines the kinetic inductance, is sensitive to evaporation parameters.

In this work, we developed a high-vacuum-compatible wireless ohmmeter with an independently controlled rotary shutter, allowing for in situ measurements of the sheet resistance of the deposited film. This method allows us to reliably realise superconducting coplanar waveguide (CPW) grAl resonators with impedance exceeding 13 kΩ, even reaching $Z = 22.3 \pm 0.3\,k\Omega$, thanks to the large sheet kinetic inductance up to $L_k = 2.7 \pm 0.1\,nH\,\square^{-1}$. These resonators offer magnetic field resilience of $B_\perp^{max} = (281 \pm 1)\,mT$ and $B_\parallel^{max} = (3.50 \pm 0.05)\,T$.

We integrate a grAl CPW resonator with a double quantum dot (DQD) device fabricated on a planar Ge/SiGe heterostructure confining holes, whose large spin-orbit interaction enables fast and all-electric control of spin qubits[41]. By using a resonator with a characteristic impedance of 7.9 kΩ and sheet kinetic inductance of 800 pH $\square^{-1}$, we demonstrate a strong hole-photon coupling with a rate of $g_c/2\pi = (566 \pm 2)\,MHz$ and cooperativity of $C = 251 \pm 8$.

## Results

### Granular Aluminium

Granular aluminium has been studied for almost 60 years[42,43] but has recently attracted increased attention as a material of interest for qubits, amplifiers and detectors[1,38]. It consists of small crystalline grains of $\approx 4\,nm$ in diameter (Fig. 1a, red and Supplementary Fig. 1)

embedded in an amorphous aluminium oxide matrix (Fig. 1a, green). The aluminium grains separated by an insulating barrier can be modelled as a network of Josephson junctions, resulting in high kinetic inductance[38]. The kinetic inductance of superconductors arises from the inertia of moving Cooper pairs and increases with normal state resistance. However, it cannot be increased infinitely; it is upper-bound by a superconducting-to-insulator phase transition (SIT). For ~ 20 nm-thin grAl films, the SIT has been measured around 5 kΩ $\square^{-1}$[44].

We prepare grAl by room temperature electron-beam evaporation of aluminium in an oxygen atmosphere at ~ 5 × 10⁻⁵ mbar pressure at a rate of 1 nm s⁻¹. The resulting resistivity of the evaporated film depends on the oxygen flow and the evaporation rate. To determine the sheet resistance, we initially used a glass sample of 10 squares and measured its resistance using a multimeter after the evaporation. The outcome of these evaporations is summarized in Fig. 1b. It is observed that the sheet resistance exponentially depends on the oxygen flow for a given evaporation rate and is not reproducible. The poor reproducibility is quantified in the blue violin plot in Fig. 1c, which shows individual evaporations of bare resonator and resonator-quantum dot samples targeting 2.5 kΩ $\square^{-1}$. Given the poor reproducibility even for nominally identical consecutive evaporations, it is challenging to systematically fabricate high-impedance CPW resonators which operate within the designed range.

To overcome this experimental difficulty, a hermetically sealed high-vacuum compatible wireless ohmmeter has been developed ("Methods" and Supplementary Fig. 2). It allows the in situ two-probe electrical measurement of the growing film resistance[45] and wireless transmission of the data. This enables the termination of the evaporation when the desired value is reached. Such a procedure yields a reproducible resistance at the expense of the uncertainty in the final film thickness. This can be decreased by equipping the ohmmeter with an independent rotary shutter, as seen in Fig. 1d. It masks three-quarters of the sample holder and is wirelessly controlled. In this manner, three test samples are evaporated before the sample of interest without having to break vacuum, thus allowing the operator to fine-tune the oxygen flow and reach the desired film properties. As shown in Fig. 1c, this procedure significantly reduces the uncertainty of

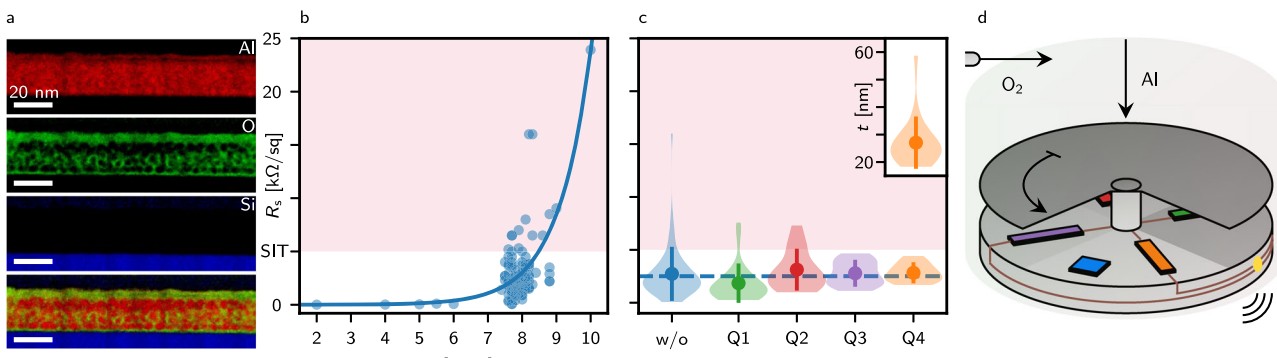

**Fig. 1 | grAl microstructure, evaporation characteristics and developed holder. a** Scanning transmission electron microscopy (STEM) electron energy loss spectroscopy (EELS) core-loss qualitative compositional maps of Al (red), O (green), and Si (blue) of a 2.5 kΩ$\square^{-1}$, 25 nm-thin grAl film deposited on a Si substrate, oriented along the [011] zone axis. The oxygen map shows an aluminium oxide matrix with embedded crystalline Al particles. The top 5 nm of the film is fully oxidised. **b** Oxygen flow dependence of the sheet resistance $R_s$ of a grAl film evaporated with a deposition rate of 1 nm s⁻¹ on a test glass piece with 10 squares as measured with a multimeter after the evaporation. The line is an exponential fit. The red colour denotes the area above the SIT. **c** Sheet resistance $R_s$ aiming at 2.5 kΩ$\square^{-1}$ for the respective quadrants of the developed sample holder and without (w/o, blue) the developed holder. The violin plots denote the distribution, points the means and bars the standard deviation. Although the median value without the holder is

2.08 kΩ$\square^{-1}$, the sheet resistances spread from 60 Ω$\square^{-1}$ to 16 kΩ$\square^{-1}$, with the mean value of 2.64 kΩ$\square^{-1}$ and a standard deviation of 2.3 kΩ$\square^{-1}$. With the holder, the resulting sheet resistance dispersion is significantly decreased at the expense of the defined film thickness, as shown in the inset. The resistances in the fourth quadrant are spread between 1.68 kΩ$\square^{-1}$ to 4.42 kΩ$\square^{-1}$, with median of 2.61 kΩ$\square^{-1}$, mean of 2.79 kΩ$\square^{-1}$ and standard deviation of 0.8 kΩ$\square^{-1}$. The median of the thicknesses 25.4 nm matches the desired value of 25 nm, while they spread around the mean of 27.1 nm with a standard deviation of 8.8 nm. **d** Schematics of the sample holder allowing in situ resistance measurements through a wireless connection. An independent rotary shutter divides the holder into four quadrants. The rectangular test pieces (green, red, purple and orange) have gold-plated ends connected via cables (brown) with the two-probe measurement circuit inside the holder through feed-throughs (gold). A square sample is shown in blue.

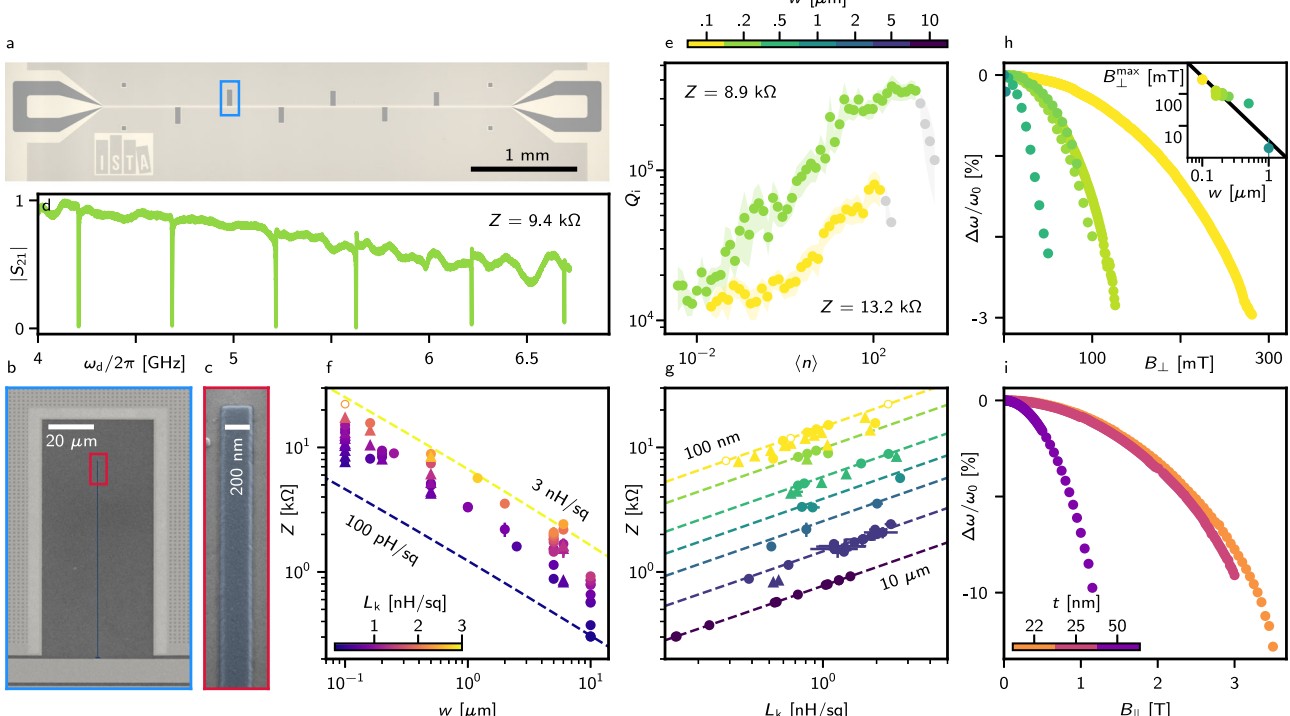

**Fig. 2 | Characteristic impedance, quality factors and magnetic field resilience of grAl CPW resonators. a** Optical microscope image of six 200 nm-wide grAl resonators side-coupled to a feedline. Due to large sheet kinetic inductance $L_k$, their length does not exceed 200 $\mu$m. The Nb ground plane around the grAl resonator is patterned with a vortex-trapping hexagonal array of circular holes with ~150 nm diamater[61]. **b, c** Scanning electron microscope image of a single resonator. **d** Magnitude $|S_{21}|$ of six bare resonators capacitively side-coupled to a common feedline. The impedance of these resonators is extracted to be $Z = (9.4 \pm 0.2)$ k$\Omega$ and the sheet kinetic inductance to be $L_k = (916 \pm 40)$ pH $\square^{-1}$. **e** Average photon number dependence of the internal quality factor for 100 nm- and 200 nm-wide resonators. The impedance is $(13.2 \pm .5)$ k$\Omega$ [$L_k = (817 \pm 62)$ pH $\square^{-1}$] and $(8.9 \pm .2)$ k$\Omega$ [$L_k = (816 \pm 28)$ pH $\square^{-1}$] respectively. The power dependence of the internal quality factor reveals behaviour suggestive of losses due to two-level systems. **f** Centre conductor width dependence of the characteristic impedance for different sheet kinetic inductances. The circles correspond to resonators fabricated on Si, while triangles to Ge/SiGe heterostructures with the Ge quantum well etched away. The empty circles indicate samples where most resonators did not resonate. The dashed lines represent the impedance calculated from (1) for 25 $\mu$m separation of the centre conductor and the ground plane and relative permittivity of 11.8 ("Methods"). The error bars are extracted from a statistical analysis of several resonators within a chip. **g** Sheet kinetic inductance dependence of the characteristic impedance for different resonator widths. **h** (**i**) Applied out-of-plane (in-plane) magnetic field dependence of the relative change in resonance frequency for different resonator widths (thickness). The inset in **h** shows the centre conductor width dependence of the maximum out-of-plane field before the internal quality factor significantly drops following $1.65\Phi_0/w^2$ (black line). The kinetic inductance $L_k$ does not influence the resonator's response to the applied magnetic field. Corresponding quality factor dependencies are shown in Supplementary Fig. 4.

both resistance, making it possible to target the desired resonance frequency range. These results demonstrate a successful implementation of our technical solution, which is also transferable to other disordered superconductors and, in general, thin-film systems.

## Granular aluminium resonators

We utilize the developed method to realize high-impedance grAl CPW $\lambda/2$ resonators. We first test the resonators on bare $\langle 100 \rangle$ silicon ($\rho = 1 - -5\,\Omega$cm) substrates. We intentionally choose low resistivity substrates as those resonators are developed for use with semiconductor heterostructures with much higher losses than highly resistive silicon or sapphire wafers[27,46,47]. The ground plane and feedline are comprised on Nb, with negligible kinetic inductance ("Methods") and employ different CPW geometries. The hanger geometry (Fig. 2a, b) is preferred since it allows testing multiple resonators per chip, as seen in Fig. 2d, as well as the precise determination of internal losses and coupling rates. Figure 2b shows a scanning electron microscope image of a single grAl resonator capacitively side-coupled to a common Nb feedline. Two strategies are employed in order to maximise the impedance: first, increasing the kinetic inductance via the sheet resistance and, second, decreasing the width of the centre conductor from 10 $\mu$m to 100 nm. Additionally, such narrow

resonators exhibit an increased out-of-plane magnetic field resilience[26], beneficial for materials with low in-plane $g$-factors[41].

In Figs. 2f, g we summarise the characteristic impedances of the fabricated resonators. Leveraging the developed wireless ohmmeter holder, we target sheet resistances of 2.5 k$\Omega\square^{-1}$ and obtain films with a kinetic inductance reaching $2.7 \pm 0.1$ nH $\square^{-1}$. The impedance scales with the kinetic inductance $L_k$ and the centre conductor width $w$ according to

$$Z(L_k, w) = \sqrt{\frac{L_\ell}{C_\ell}} = \sqrt{\frac{L_\ell^k + L_\ell^g}{C_\ell}} = \sqrt{\frac{\frac{L_k}{w} + L_\ell^g(w)}{C_\ell^g(w)}}, \quad (1)$$

with $L_\ell$ ($C_\ell$) the inductance (capacitance) per length. For a superconducting resonator, the inductance per length $L_\ell = L_\ell^k + L_\ell^g$ is composed of a geometric $L_\ell^g$ and a kinetic $L_\ell^k = L_k/w$ contribution ("Methods"). The highest impedance is reached for widths below 200 nm. However, we observe a limitation with three 100 nm-wide samples, with most devices failing to respond to microwave excitation. These samples are highlighted with empty circles in Figs. 2f, g. The highest characteristic impedance observed for such a sample is $(22.3 \pm 0.3)$ k$\Omega$. While the reason remains unclear, we speculate that

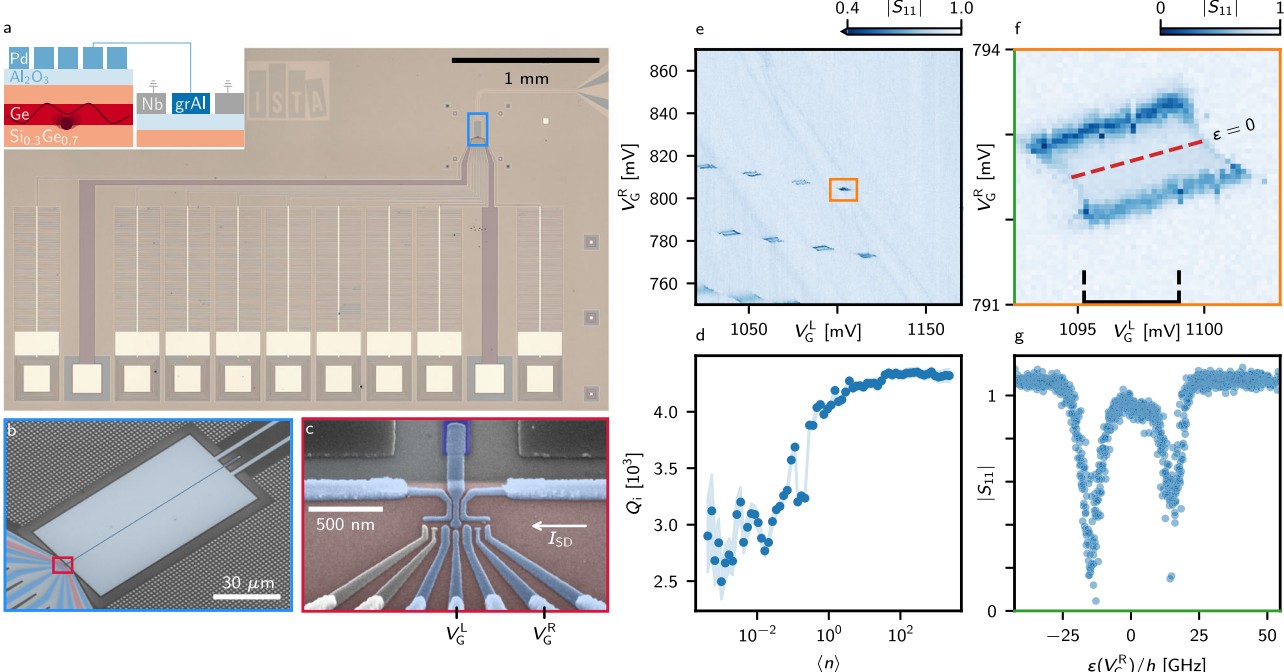

**Fig. 3 | Integration of a 7.9 kΩ grAl CPW resonator with germanium quantum dots. a** Optical microscope image of a quantum dot cQED device implementing a grAl reflection resonator with a triple quantum dot electrostatically defined by nine gates. In this experiment, it was operated as a DQD, as indicated in the schematic cross-section of the device in the inset, which illustrates a hole delocalized in a DQD potential. **b** Scanning electron microscope image showing detail of the region highlighted with blue in **a**. The resonator grAl centre conductor (dark blue), which is designed to be ≈ 111 μm long and 200 nm-wide, was evaporated with 25 nm thickness. The red region highlights the mesa. **c** Scanning electron microscope image showing detail of the region highlighted with red in **b**. The gates coloured in grey were grounded throughout the experiments. **d** Average photon

number dependence of the internal quality factor for the resonator implemented in a quantum dot cQED device with grounded gates and a Ge hole gas in the vicinity of the tip of the resonator. The impedance of the resonators is ≈ 7.9 kΩ. **e** Stability diagram of the right DQD sensed by the reflected amplitude at the drive frequency ≈ 2 MHz below the resonance frequency as a function of the two plunger gates defining the right DQD. The number of confined holes is unknown. **f** A single interdot transition as seen by the reflected amplitude. The black dashed lines highlight the range of $V_G^L$, which is used for further analysis. **g** Line cut taken along $V_G^R$ from the indicated range, centred around zero DQD detuning representing the average of ~ 20 line cuts, showing the detuning dependence of the reflection amplitude.

for smallest widths, the evaporator background pressure could play a role. In contrast, other 100 nm-wide samples demonstrate reliable performance. Specifically, five samples comprising 20 resonators with impedance ranging from 13 kΩ to 17.3 kΩ show a 100% yield. These results demonstrate that the impedance of the grAl CPW resonators can reproducibly exceed 13 kΩ, strongly enhancing quantum fluctuations of the voltage.

We next measure the complex frequency response with respect to the driving power and evaluate the quality factors as a function of the photon number ("Methods"). Figure 2e displays a power sweep of two resonators with widths of 100 nm and 200 nm. The resonators retain a quality factor above $10^4$ in the single photon regime, even with impedance exceeding 13 kΩ. We conclude that high impedance grAl resonators exhibit losses that would not limit initial cQED experiments, with loss rates in the order of MHz, which is comparable with the state-of-the-art charge/spin qubits decoherence rates[15].

To evaluate the potential of the developed CPW resonators for spin-photon coupling experiments, we investigate their magnetic field behaviour. Since the g-factor of holes is highly anisotropic, with rather small in-plane and large out-of-plane g-factors[48], we probe their behaviour for both magnetic field directions. In Fig. 2h, i (Supplementary Fig. 4), the relative frequency shift (the internal quality factors) as a function of the applied magnetic fields for resonators with hundreds of photons is plotted. The superconducting depairing parabolically lowers the resonance frequency due to increased kinetic inductance, stemming from the decreasing superconducting gap. Furthermore, the magnetic field induces losses by introducing

Abrikosov vortices in the film, whose dynamics contribute to microwave losses and resonator instability. The creation of vortices is suppressed with narrow centre conductors. Thus, the maximal out-of-plane field the resonators withstand before the deterioration $B_\perp^{max}$ increases with decreasing widths, and it approximately follows $B_\perp^{max} = 1.65\Phi_0/d^2$ [49], where $\Phi_0 = h/(2e)$ is the superconducting flux quantum and $d$ is the dimension perpendicular to the applied field (inset Fig. 2g). We note that for 100 nm-wide resonators, the magnetic field resilience reaches $B_\perp^{max} = (281 \pm 1)$ mT out-of-plane. Moreover, since the creation of vortices is also suppressed with thin films in the in-plane field, resonators with thickness below 25 nm withstand 3 T, the thinnest reaching $B_\parallel^{max} = (3.50 \pm 0.05)$ T.

In the context of spin qubits, the demonstrated combination of high impedance and magnetic field resilience makes grAl a strong candidate for a long-range coherent coupler. For silicon-based spin qubits (g-factor of 2), the spin transition can be easily brought to the 4–8 GHz range where microwave resonators are routinely operated. Moreover, such magnetic field resilience allows one to readily implement grAl resonators with spin qubits hosted in materials with highly anisotropic g-factors, such as Ge. In this case, the spin qubit could be operated in an arbitrary magnetic field orientation, an important feature of operating the qubit in the parameter range where its coherence is maximized[50].

### Strong charge-photon coupling

The strong coupling condition requires that the coherent coupling strength $g_c$ exceeds both the charge (γ) and the resonator (κ) decay

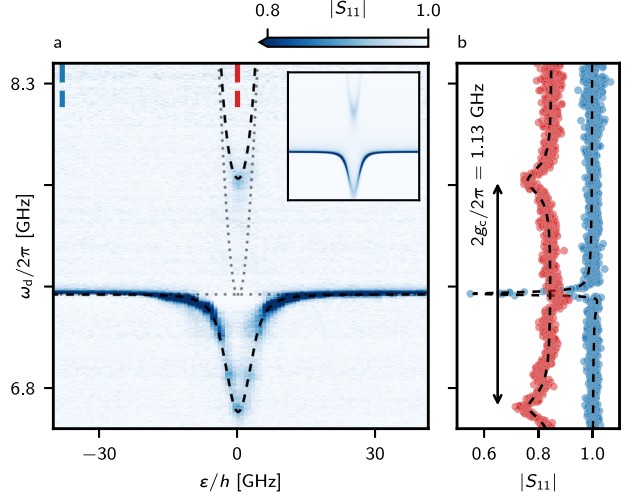

**Fig. 4 | Strong charge-photon coupling. a** Resonator microwave spectroscopy as a function of drive frequency $\omega_d$ and DQD detuning $\varepsilon$. Two well-isolated dressed states are observed. At $\varepsilon = 0$ and with the charge transition set resonant with the resonator, the vacuum Rabi splitting is observed, demonstrating the strong coupling regime. The dotted lines show the bare resonator frequency and energies of the bare charge transition, while the dashed lines show the dressed transitions, with frequencies extracted from the line cut fits. The inset shows a theoretical reproduction of the experimental data ("Methods"). **b** Magnitude line cuts along the frequency axes, taken at the detunings indicated by the dashed blue and red labels. A fit to the line-cut at zero detuning in the right panels gives $(g_c/2\pi, \gamma/2\pi, t_c/h) = (566, 297, 3626) \pm (2, 9, 2)$ MHz, yielding a cooperativity of $251 \pm 8$. The red curve is offset by 0.15 for clarity.

rates. To coherently couple a charge with a photon, we integrate a grAl resonator with a DQD formed in a Ge/SiGe heterostructure[51] (Figs. 3a to c). We opted for a reflection geometry, which improves the signal-to-noise by a factor of two relative to the hanger geometry, where the signal can leak back to the input port[52]. The resonance frequency is measured to be 7.262 GHz, yielding a characteristic impedance of 7.9 kΩ and sheet kinetic inductance of 800 pH □⁻¹. The resonator is designed to be over-coupled, so the coupling quality factor $Q_c$ is $(534.8 \pm 4.7)$, while the internal quality factor $Q_i$ is $(2000.4 \pm 84.7)$ with $\approx 0.2$ average photons in the resonator and a DQD formed, leading to the resonator field decay rate of $\kappa/2\pi = (17.2 \pm 0.3)$ MHz. The quality factors are significantly smaller compared to bare resonators (Fig. 3d). As described in Ref. 47, low-impedance resonators fabricated on Ge-rich substrates featuring etched quantum wells exhibit reduced internal quality factors of $\sim 10^4$ in the single photon regime[47]. Moreover, the metallic gates for defining the quantum dots will cause significant microwave leakage, even more pronounced for high characteristic impedance[27].

The triple quantum dot array is operated as a DQD formed on the right side, with the two left-most gates grounded. The DQD is first tuned by measuring a DC current. As observed in the stability diagram in Fig. 3e, the resonator is dispersively shifted at the interdot transitions, i.e. when the DQD dipole moment becomes sizeable, allowing for direct microwave readout of individual interdot transitions (Fig. 3f). Line cuts along the right gate voltage (Fig. 3g) translated to detuning through lever arms ("Methods" and Supplementary Fig. 3) exhibit a sharp response in the complex resonator signal due to the interaction with the DQD. The clear dispersive interaction between the DQD and the resonator already hints towards a significant charge-photon interaction strength. However, this type of measurement does not provide sufficient information to extract the system parameters reliably[53].

In contrast, sweeping the drive frequency while stepping the DQD detuning reveals the spectrum of the hybridized states (Fig. 4a), allowing for accurate characterization of the system. Thus, we focus on the interdot transition shown in Fig. 3f and we probe the system by reflection spectroscopy. A parabolic spectrum is expected for a DQD occupied with an odd number of holes. A clear avoided crossing is observed when the parabolic DQD charge transition frequency is tuned resonant with the resonator ($2t_c/h \approx \omega_r/2\pi$, where $t_c$ is the DQD tunnel coupling), as seen in Fig. 4a. Since the strong coupling condition $g_c > (\kappa, \gamma)$ is met, two well-resolved dressed states separated by $2g_c/2\pi$ emerge. The resonant vacuum Rabi splitting is observed (Fig. 4b), which constitutes the experimental evidence of the strong coupling regime. From Fig. 4b we extract $g_c/2\pi$ of $(566 \pm 2)$ MHz, with $\gamma/2\pi$ of $(297 \pm 9)$ MHz, demonstrating strong charge-photon coupling with a cooperativity $C = 4g_c^2/(\gamma\kappa)$ of $251 \pm 8$[54]. The coupling strength agrees with the expected value $g_c^e/2\pi = e\beta(\omega_r/2\pi)\sqrt{Z/\pi\hbar}/2 \approx 685$ MHz within less than 20%, with $e$ being the elementary charge, given a differential lever arm $\beta = 0.241$ ("Methods"), $\omega_r/2\pi = 7.27$ GHz and $Z = 7.9$ kΩ.

## Discussion

We have developed a wireless ohmmeter which enables us to in situ define the sheet resistance of grAl CPWs, allowing for reproducible fabrication of grAl resonators with characteristic impedance exceeding 13 kΩ, even reaching 22 kΩ. The reported values are more than four times higher than the state-of-the-art for semiconductor circuit quantum electrodynamics experiments[17].

Furthermore, we have shown that grAl resonators remain low-loss and are resilient to magnetic field up to 3.5 T in-plane and 281 mT out-of-plane. By integrating a 7.9 kΩ grAl resonator with a DQD formed in planar Ge we demonstrate strong hole-photon coupling with a rate of $g_c/2\pi = (566 \pm 2)$ MHz. The demonstrated charge-photon coupling strength and magnetic field resilience of the grAl resonators combined with the strong spin-orbit coupling opens the path towards strong spin-photon coupling in planar Ge[41]. Even for modest spin-orbit interaction, similar to that created by a gradient magnetic field of a micromagnet employed in Si[23,24], spin-photon coupling rates could exceed 100 MHz. Such high rates suggest that two-qubit operations between distant hole spins with high fidelities are within reach[25,55,56].

In conclusion, we have proposed controllably deposited grAl as an alternative to Josephson junction arrays and atomically disordered superconductors for semiconductor-superconductor circuit quantum electrodynamics experiments. Its key advantages lie in being high impedance, magnetic field resilient, lift-off compatible and low-loss. By further optimising the design to reach $\beta \approx 0.25$ and using an impedance of 16 kΩ, charge-photon coupling strengths above 1 GHz are within reach. Moreover, the impedance can be increased beyond these values by meandering the grAl resonator and utilizing the mutual geometric inductance[3,34]. Using a grAl resonator, given the state-of-the-art quantum dot technologies, spin-photon coupling could enter the ultra-strong coupling regime, following in the footsteps of the superconducting qubits and opening an avenue towards unexplored physics[57] and advanced quantum information processing applications.

## Methods

### Vacuum-compatible wireless ohmmeter

The wireless ohmmeter was designed to fit the MEB550S Plassys HV electron-beam evaporator, equipped with a motorized evaporation stage. The stage can tilt between the loading and evaporation positions and rotate along the evaporation axis to ensure a uniform distribution of the deposited thin film. The ohmmeter was designed to communicate wirelessly with the handheld device outside the vacuum chamber so as not to restrict the movement of the rotary stage. The designed device uses the surface of the ohmmeter enclosure lid as a

mounting surface for the samples and measuring probes. The ohmmeter circuit is placed inside the hermetically sealed enclosure, and the measuring electrodes are connected to the ohmmeter through hermetically sealed electrical cable feedthroughs. The wireless in situ resistance measurement enables the user to monitor the sheet resistance of the deposited film and to interrupt the process when the desired resistance is reached.

An independently controlled rotary shutter is magnetically coupled to a servo motor enclosed inside the holder, as magnetic coupling is a straightforward method for transferring motion from a vacuum-non-compatible motor to the high vacuum side.

The battery-powered wireless ohmmeter includes components to precisely measure the growing thin film resistance and transmit the measured values to the recording device outside the vacuum chamber. The design is based on the ATmega2560 microcontroller to communicate with the rest of the components, which include a 2.4GHz wireless transceiver, a barometer to record the temperature and pressure inside the hermetic enclosure for detecting potential pressure leaks, a servo motor to move the magnetically coupled shutter, and a precision analogue-to-digital converter to measure resistance using the constant voltage method. The highest sensitivity is achieved by setting the range resistor closest to the measured value. Since the resistance of the deposited grAl spans several orders of magnitude, we use a set of four different resistors selected by switching mechanical relays to cover a large measurement range.

## Sample fabrication

The bare resonators are fabricated on $\langle 100 \rangle$ silicon substrates ($\rho \approx 1\text{–}5\,\Omega$cm). First, 5/60 nm Cr/Au alignment markers are patterned using 100 keV electron beam lithography (EBL), high-vacuum (HV) electron beam (e-beam) evaporation and lift-off process. Next, a 20 nm Nb feedline and ground plane are defined using EBL, ultra-high vacuum (UHV) e-beam evaporation and lift-off. The grAl is not employed in this layer since achieving 50 $\Omega$ matching with a kinetic inductance of 2 nH $\square^{-1}$ would require a 1 mm-wide feedline separated from the ground plane by 100 nm. Finally, the ~ 25 nm grAl centre conductors are patterned with EBL, HV e-beam evaporation in an oxygen atmosphere and lift-off. After the evaporation, the film is subject to a 5 min static post-oxidation.

The quantum dot cQED devices are fabricated on a Ge/SiGe heterostructure grown by low-energy plasma-enhanced chemical vapour deposition with forward grading[51]. The two-dimensional hole gas is self-accumulated in the Ge quantum well $\approx 20$ nm below the surface.

The process is as follows: First, the ohmic contacts are patterned with EBL. Before the deposition of 60 nm of Pt at an angle of 5°, a few nanometres of native oxide and SiGe spacer are removed by argon bombardment. Next, the hole gas is dry-etched with $SF_6$-$O_2$-$CHF_3$ reactive ion etching everywhere except a small ~ 60 nm high mesa area extending to two bonding pads to form a conductive channel. Subsequently, the native $SiO_2$ is removed by a 10 s dip in buffered HF before the gate oxide is deposited. The oxide is a ~ 10 nm atomic-layer-deposited aluminium oxide ($Al_2O_3$) film grown at 300 °C. Next, a 20 nm Nb feedline and ground plane are fabricated with EBL, UHV evaporation and lift-off. The separation of the grAl centre conductor from the Nb ground plane is 25 $\mu$m for resonators with a width less than 1 $\mu$m. Further retraction of the ground plane would not increase the impedance appreciably. The single-layer top gates for defining the QDs are patterned in three different steps of EBL, evaporation and lift-off. The barrier and plunger gates are defined separately close to the edge of the mesa with Ti/Pd 3/20 nm. An additional Ti/Pd 3/97 nm-thick gate metal layer defines microwave filters and the bonding pads, connects to the previously defined gates and overcomes the slanted mesa edge. Finally, the grAl centre conductors are defined as in the bare resonator samples.

## Measurement setup

The reported low-temperature measurements were performed in a cryogen-free dilution refrigerator with a base temperature of 10 mK. The sample was mounted on a custom-printed circuit board thermally anchored to the mixing chamber of the cryostat. The electrical connections were made via wire bonding. The schematic of the measurement setup is shown in Supplementary Fig. 5.

## Impedance evaluation

Since the characteristic impedance cannot be measured directly, we extract it from the resonance frequency. $C_\ell = C_\ell^g(w)$ and $L_\ell^g(w)$ in (1) are solely given by the design parameters (the centre conductor width $w$, the separation between the ground plane and the centre conductor $s$ and the relative permittivity of the substrate $\varepsilon_r$) and can be analytically calculated. We extract the characteristic impedance from the resonance frequency, which is given as

$$\omega_r = \frac{1}{\sqrt{L_{\lambda/2}\left(C_{\lambda/2} + C_c\right)}}, \tag{2}$$

where $L_{\lambda/2} = (2\ell/\pi^2)L_\ell$ and $C_{\lambda/2} = (\ell/2)C_\ell$ are lumped-element equivalents of the distributed parameters of the resonator. For longer, lower-impedance resonators, the coupling capacitance $C_c$ is usually dominated by $C_{\lambda/2}$ and can be neglected. In the case of our compact high-impedance, they are of the same order of magnitude, and both have to be considered. We simulate the coupling capacitance using COMSOL electrostatic simulations. We measure the complex frequency response of a resonator and fit the signal to evaluate the resonance frequency. We calculate the sheet kinetic inductance $L_k$ by numerically solving (2) with the simulated coupling capacitance.

## Fitting

The Heisenberg-Langevin equation of motion for a bare hanger resonator in the rotating frame is

$$\dot{\hat{a}} = -\frac{i}{\hbar}[\hat{a}, \hbar(\omega_r - \omega_d)\hat{a}^\dagger \hat{a}] - \frac{\kappa}{2}\hat{a} - \sqrt{\frac{\kappa_c}{2}}\hat{b}_{in}$$
$$= -i(\omega_r - \omega_d)\hat{a} - \frac{\kappa}{2}\hat{a} - \sqrt{\frac{\kappa_c}{2}}\hat{b}_{in}, \tag{3}$$

where $\hat{a}$ is the resonator field, $\hat{b}_{in}$ is the input field and $\kappa_c$ is the resonator coupling rate. For the steady state ($\dot{\hat{a}} = 0$), it becomes

$$\hat{a} = \frac{-\sqrt{\frac{\kappa_c}{2}}}{i(\omega_r - \omega_d) + \frac{\kappa}{2}}\hat{b}_{in}. \tag{4}$$

Using the input-output relation $\hat{b}_{out} = \hat{b}_{in} + \sqrt{\frac{\kappa_c}{2}}\hat{a}$, where $\hat{b}_{out}$ is the output field, yields the complex transmission of a bare hanger-type resonator

$$S_{21}(\omega_d) = \frac{\langle \hat{b}_{out} \rangle}{\langle \hat{b}_{in} \rangle} = 1 - \frac{\kappa_c e^{i\phi}}{\kappa + 2i(\omega_r - \omega_d)}, \tag{5}$$

with an inserted term $e^{i\phi}$ accounting for the impedance mismatch. Before fitting, the feedline transmission $S_{21}$ is first normalized by a background trace to remove the standing wave pattern obtained from a scan with a shifted resonance frequency, either due to a magnetic field or a DQD-induced dispersive shift. The average intraresonator photon number at resonance was calculated as

$$\langle n \rangle = \frac{2}{\hbar\omega_r}\frac{\kappa_c}{\kappa^2}P_d, \tag{6}$$

where $P_d$ is the drive power after accounting for input attenuation. In the case of a single port resonator, (6) gains an extra factor of two.

For a DQD occupied with an odd number of holes, the cavity-DQD Hamiltonian in the rotating frame and rotating wave approximation can be described by

$$\hat{H}/\hbar = (\omega_r - \omega_d)\hat{a}^\dagger \hat{a} + \frac{(\omega_q - \omega_d)}{2}\hat{\sigma}_z + g_{eff}\left(\hat{a}^\dagger \hat{\sigma}_- + \hat{a}\hat{\sigma}_+\right), \quad (7)$$

where $\omega_d$ is the drive frequency, $\omega_r$ is the resonance frequency, $\omega_q = \sqrt{\varepsilon^2 + 4t_c^2}/h$ is the charge transition frequency with $\varepsilon$ being the DQD detuning and $t_c$ the tunnel coupling. $g_{eff} = g_c \frac{2t_c}{\hbar\omega_q}$ is the effective charge-photon coupling strength, $\hat{a}^\dagger$ ($\hat{a}$) is the photon creation (annihilation) operator and $\hat{\sigma}_z$ is the Pauli $z$ matrix and $\hat{\sigma}_-$ ($\hat{\sigma}_+$) is the qubit lowering (raising) operator.

The Heisenberg-Langevin equations of motion for a system described by (7) are[58]

$$\begin{aligned}\dot{\hat{a}} &= -\frac{i}{\hbar}[\hat{a}, \hat{H}] - \frac{\kappa}{2}\hat{a} - \sqrt{\kappa_c}\hat{b}_{in} \\ &= -i(\omega_r - \omega_d)\hat{a} - \frac{\kappa}{2}\hat{a} - \sqrt{\kappa_c}\hat{b}_{in} - ig_{eff}\hat{\sigma}_-,\end{aligned} \quad (8)$$

$$\begin{aligned}\dot{\hat{\sigma}}_- &= -\frac{i}{\hbar}[\hat{\sigma}_-, \hat{H}] - \frac{\gamma}{2}\hat{\sigma}_- \\ &= -i(\omega_q - \omega_d)\hat{\sigma}_- - \frac{\gamma}{2}\hat{\sigma}_- - ig_{eff}\hat{a}.\end{aligned} \quad (9)$$

For the steady state ($\dot{\hat{\sigma}}_- = 0$), (9) becomes

$$\hat{\sigma}_- = \frac{g_{eff}}{-(\omega_q - \omega_d) + i\gamma/2}\hat{a}, \quad (10)$$

which inserted in (8) with $\dot{\hat{a}} = 0$ gives

$$\hat{a} = \frac{-\sqrt{\kappa_c}}{i(\omega_r - \omega_d) + \frac{\kappa}{2} + \frac{ig_{eff}^2}{-(\omega_q - \omega_d) + i\gamma/2}}\hat{b}_{in}. \quad (11)$$

Using $\hat{b}_{out} = \hat{b}_{in} + \sqrt{\kappa_c}\hat{a}$ yields the complex reflection of a single-port resonator coupled to a DQD charge transition

$$S_{11}(\omega_d) = \frac{\langle\hat{b}_{out}\rangle}{\langle\hat{b}_{in}\rangle} = 1 - \frac{2\kappa_c e^{i\phi}}{\kappa + 2i(\omega_r - \omega_d) + \frac{2ig_{eff}^2}{-(\omega_q - \omega_d) + i\gamma/2}}. \quad (12)$$

**Lever arms**

The translation between the plunger gate voltages and DQD detuning is given by[59]

$$\varepsilon = \mu_L - \mu_R = (\alpha_{LL} - \alpha_{RL})\Delta V_G^L - (\alpha_{RR} - \alpha_{LR})\Delta V_G^R, \quad (13)$$

where $\mu_L$ ($\mu_R$) is the electrochemical potential of the left (right) quantum dot, and $\alpha_{ij}$ is the lever arm of the gate $i$ on dot $j$ ($i, j \subset \{L,R\}$). The plunger lever arms are extracted from bias triangles of the DQD, shown in Supplementary Fig. 3.

To quantify the capacitive effect of the resonator gate on the double quantum dot (DQD) with a differential lever arm, a test sample was fabricated. A DC voltage was connected to the gate mimicking the resonator extension, a DQD was formed, and lever arms were extracted from Coulomb diamonds. The lever arm of the 'resonator' gate to the underlying quantum dot was $\alpha_{ResQD} = .265 \pm 0.019$. Since the charge-photon coupling is given by the electrochemical potential difference between the dots induced by the resonator, the $\alpha_{ResQD}$ lever arm needs to be corrected for the effect on the side dot. The cross-lever arm, between the resonator and the side quantum dot, was extracted to be $\alpha_{ResSD} = .024$. The differential lever arm is thus estimated to be $\beta = \alpha_{ResQD} - \alpha_{ResSD} = .241 \pm 0.019$. The discrepancy between the estimated and measured charge-photon coupling rate could be explained by the lever arm dependence on the number of confined holes and DQD tuning.

## Data availability

All experimental data included in this work are available via ISTA Research Data Repository at https://doi.org/10.15479/AT:ISTA:18886.

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

## Acknowledgements

We acknowledge Franco De Palma, Mahya Khorramshahi, Fabian Oppliger, Thomas Reisinger, Pasquale Scarlino and Xiao Xue for helpful discussions. We thank Simon Robson for proofreading the manuscript. This research was supported by the Scientific Service Units of ISTA through resources provided by the MIBA Machine Shop and the Nanofabrication facility. This research and related results were made possible with the support of the NOMIS Foundation and the HORIZON-RIA 101069515 project. This research was funded in whole or in part by the Austrian Science Fund (FWF) https://doi.org/10.55776/P32235, https://doi.org/10.55776/I5060 and https://doi.org/10.55776/P36507. For Open Access purposes, the author has applied a CC BY public copyright license to any author accepted manuscript version arising from this submission. M.J. acknowledges funding from FellowQUTE 2024-01. K.R. acknowledges funding from the European Union's Horizon 2020 research and innovation program under the Marie Skłodowska-Curie Grant Agreement No. 101034413. I.M.P. acknowledges funding from the Deutsche Forschungsgemeinschaft (DFG - German Research

Foundation) under project number 450396347 (GeHoldeQED). ICN2 acknowledges funding from Generalitat de Catalunya 2021SGR00457. We acknowledge support from CSIC Interdisciplinary Thematic Platform (PTI+) on Quantum Technologies (PTI-QTEP+). This research work has been funded by the European Commission - NextGenerationEU (Regulation EU 2020/2094), through CSIC's Quantum Technologies Platform (QTEP). ICN2 is supported by the Severo Ochoa programme from Spanish MCIN/AEI (Grant No.: CEX2021-001214-S) and is funded by the CERCA Programme/Generalitat de Catalunya. Part of the present work has been performed in the framework of Universitat Autònoma de Barcelona Materials Science PhD programme. AGM has received funding from Grant RYC2021-033479-I funded by MCIN/AEI/10.13039/501100011033 and by European Union NextGenerationEU/PRTR. M.B. acknowledges support from SUR Generalitat de Catalunya and the EU Social Fund; project ref. 2020 FI 00103. The authors acknowledge the use of instrumentation and the technical advice provided by the Joint Electron Microscopy Centre at ALBA (JEMCA). ICN2 acknowledges funding from Grant IU16-014206 (METCAM-FIB) funded by the European Union through the European Regional Development Fund (ERDF), with the support of the Ministry of Research and Universities, Generalitat de Catalunya. ICN2 is a founding member of e-DREAM[60].

## Author contributions

M.J. developed the fabrication recipes, designed the resonator and quantum dot cQED devices and analysed data. K.R. developed and characterized the grAl evaporation process for the wireless ohmmeter with the help of M.J. M.J., K.R. and C.B.E. fabricated samples and performed experiments. A.B. and T.A. conceived, designed and built the wireless ohmmeter. M.J. and O.S. developed the microwave technology for the Ge/SiGe heterostructures. S.C., D.C. and G.I. designed and grew the Ge/SiGe heterostructure. M.B., A.G.M. and J.A. performed the atomic resolution scanning transmission electron microscopy structural and electron energy-loss spectroscopy compositional-related characterization. M.J., K.R. and G.K. contributed to discussions and the preparation of the manuscript, with input from the rest of the authors. G.K. and I.P. initiated, and G.K. supervised the project.

## Competing interests

The authors declare no competing interests.

## Additional information

**Peer review information** : *Nature Communications* thanks Marta Pita-Vidal, and the other, anonymous, reviewers for their contribution to the peer review of this work. A peer review file is available.

