## [Transparent Peer Review file · Nature Communications]

Strong Charge-Photon Coupling in Planar Germanium Enabled by Granular Aluminium Superinductors

Corresponding Author: Mr Marián Janík

Version 0:

Reviewer comments:

Reviewer #1

(Remarks to the Author)

Summary:

The paper discusses the integration of granular aluminum (grAl) resonators with germanium-based double quantum dots to achieve strong charge-photon coupling. This research is significant for advancing long distance couplings in quantum dot based systems. Granular aluminum, known for its high kinetic inductance, is used to create superconducting coplanar waveguide resonators with high impedance, facilitating strong coupling rates. Key achievements include the development of a high-vacuum-compatible wireless ohmmeter for in situ measurement, allowing precise control over the film's kinetic inductance during deposition. This innovation addresses the challenge of poor reproducibility in high-impedance films. The study demonstrates a strong charge-photon coupling rate of approximately 566 MHz and a cooperativity of 251, using a resonator with a characteristic impedance of 7.9 k Ω . The results pave the way for novel qubit designs and high-fidelity, long-distance two-qubit gates, crucial for scalable quantum computing.

The authors have achieved very nice results addressing a key challenge in using high kinetic inductance films for hybrid systems and shown it can be integrated with SiGe based quantum dots with a strong charge-photon coupling. I believe their results are suitable for publication in Nature Communications when concerns below are addressed:

- 1) How does the kinetic inductance compare to measured R_s , does it follow Mattis-Bardeen formula. If so where does the films sheet inductance of the order $\sim nH/square$ fall on the SIT of the grAl films. Can you comment on SIT of the grAl films processed using this particular method and process.
- 2) The authors reference [44] as the measured SIT of their grAl, however, the SIT can be highly substrate-dependt and process dependent: one could imagine different screening effects from a low resistivity substrate. I ask the authors to comment on the SIT specifically of their processes. In particular the reference [44] studied grAl on highly resistive substrates such as sapphire and different surface morphology. How does surface roughness and substrate resistivity impact the grAl properties and its SIT. It seems important to this work and future applications to determine the exact SIT for this work. I recommend additional data or thorough discussion on this.
- 3) Do the authors measure the $R_s(T)$ and do they see direct SIT or possibly interrupted by dissipative phases, such as anomalous metal phases. Important for determining origin of dissipation in their system. It seems that sufficient transport characterizations are missing to evaluate the SIT of their grAl.
- 4) Comment: what is the expected sheet kinetic inductance to possibly be achieved using their grAl process.
- 5) Figure 1 comment: what is the exponential fit, I see no comments or details about the fit. Is the red shaded region above SIT mark inferred from ref [44] or measured?
- 6) Quality factors are 10^4 , which is 'high', however, low compared to other cQED systems and microwave resonator circuits. Can the authors comment on the enhanced loss. Did the authors compare Qs on an insulating substrates.
- 7) Figure 2: in particular the authors see Q change orders of magnitude with power suggesting TLS or quasiparticles. Seems important to comment on intrinsic losses in the resonators and the impact on coherent photon-charge couplings.
- 8) Figure 2 (f): it is very difficult to make out triangles and circles in the plot. Is Si substrate low resistivity substrates?
- 9) Is the lever arm value 0.2 extracted experimentally or a fit parameter or simulation parameter. This should be clarified bottom p. 5
- 10) The authors claim high internal Q at finite field, but the field data is taken at high photon number ~ 100 . How does the

single photon Q change with applied fields. This should be important for applications in QD devices that need large, applied fields to operate.

Reviewer #2

(Remarks to the Author)
[Review attached as pdf]

Reviewer #3

(Remarks to the Author)

Referee report on the manuscript "Strong Charge-Photon Coupling in Planar Germanium Enabled by Granular Aluminium Superinductors" by Janík et al.

In this work the authors present a technological development which enables the reproducible and controllable fabrication of high impedance superconducting microwave resonators, in this case using granular aluminum. The authors present a complete characterization of their method and demonstrate a strong coupling between a high impedance resonator and a charge qubit realized in a Ge hole gas. As the authors point-out, high kinetic inductance superconductors are interesting for the realization of qubits, amplifiers, photon detectors, and for the fabrication of high-impedance resonators which may couple strongly to spin and charge qubits, or other mesoscopic systems. Therefore, I totally agree with the authors on the fact that having a method to fabricate reproducible thin films of disordered superconductors with a given sheet resistance is highly desirable. The method presented in this work is very nice and rather simple (a wireless ohmmeter), in this case applied to the evaporation of granular aluminum films, but can be extended to other materials. I find that the characterization of the microwave resonators fabricated with granular Al is rather complete (and useful), and obtaining such large impedances, larger than the resistance quantum, is a clear demonstration of the platform. I think the results give a fair account of the main advantages of the fabricated resonators in term of kinetic inductance, impedance, internal quality factor and magnetic field dependence. The results on the photon-charge coupling are also impressive although the experimental evidence given in the work is scarce, I would have preferred to show different regions in the stability diagram, discuss more in detail the signal, etc. I give in the following a list of comments and questions. I recommend publication in Nature Communications after answering my criticism.

Comments and questions:

1- In the abstract: "However, fully exploiting the potential of disordered or granular superconductors is challenging, as their inductance and, therefore, impedance at high values are difficult to control." I find the wording a bit weird. "Impedance at high value hard to control". I think it is not clear at this point what "control" means.

Following: "Here we have integrated a granular aluminium resonator, having a characteristic impedance exceeding the resistance quantum, with a germanium double quantum dot and demonstrate strong charge-photon coupling with a rate of $gc/2\pi = (566 \pm 2)$ MHz. This was achieved due to the realisation of a wireless ohmmeter, which allows in situ measurements during film deposition and, therefore, control of the kinetic inductance of granular aluminium films. Reproducible fabrication of circuits with impedances (inductances) exceeding 13 k Ω (1 nH per square) is now possible." I think, it would be better in the other order, explain first the problem, the technological solution and then the application. The wording "and therefore," is used twice and could be avoided by reorganizing the sentences better.

2- Lines 29 and 31, achieving and achieved are used one after the other. Lines 27-29 long-distance two-qubit gates is repeated and this could be avoided.

3- Lines 39-40 "Outside the field of hybrid devices" it is not clear at this point what does hybrid refers to.

4- Line 39, it is written "extended Fig". This repeat in many places in the text.

5- Line 81, on Granular Al, I love Fig 1a and extended fig E1. In particular this later I find it incredible.

6- Line 110. What does "cQED samples" mean?

7- Line 121, "Such a procedure yields a reproducible resistance at the expense of the uncertainty in the final film thickness." As I understand this makes sense, because the feedback is on the resistance. Tuning with three test samples can reduce the error, but there is no control on the thickness. The sentence in the text is "As shown in Fig. 1c, this procedure significantly reduces the uncertainty of both resistance and thickness, making it possible to target the desired resonance frequency range." But in the figure there is a large dispersion in the thickness, which is also mentioned in the caption: "With the holder, the resulting sheet resistance dispersion is significantly decreased at the expense of the defined film thickness, as shown in the inset." All in all, there is some inconsistency. In addition, Fig. 1c is not described in the main text, where the dispersion among the quadrants come from?

8- I find Fig.2 a bit hard to read. There are three different color scales which are used for different things (Lk, w, thickness) in some cases with the same colors. I would suggest a strong revision of this figure to make it more friendly with the reader. Maybe separating with subtitles, or using disjoint color scales...

9- I would appreciate a more precise description of the difference between the devices. In the text you only mention: "Two strategies are employed in order to maximise the impedance: first, increasing the kinetic inductance via the sheet resistance and, second, decreasing the width of the center conductor from 10 μm to 100 nm." There are six resonators per chip. I guess all of them have the same sheet resistance and thickness. Do they have different central conductor width? Or different length? What does make the frequency difference? How large is the dispersion of the resulting Lk comparing the resonators of the same chip? What about the quality factor dispersion within a chip?

10- In the description of the resonators, we don't see the coupling to the transmission line. How big it is?

11- Fig. 2d is never called in the text.

12- In the caption of Fig 2 : "Optical and b, c scanning electron microscope images of six 200nm-wide grAl resonators side-coupled to a feedline", b and c correspond to a single resonator.

13- Line 159, for the description of Figs. 2f, g, it is not clear in the main text how the value of the Lk is obtained. The

explanation is given in the methods, but since Eq 1 is there and since the following discussion is on L_k , it would be much better to explain how this value is extracted from the measurement of the resonances. Additionally, have the authors tried to fit the participation ratio from the temperature dependence of the resonance frequency?

14- Line 170, "However, we observe a limitation with three 100 nm-wide samples, with most devices failing to respond to microwave excitation." Why is this? I don't understand. And then... "Nevertheless, other 100 nm-wide samples demonstrate reliable performance". This is all vague.

15- Line 182, when discussion quality factor. Have you observed that quality factor decreases with kinetic inductance (sheet resistance)? Have you observed that quality factor is affected by film thickness? I guess the gray points in Fig 2e correspond to the non-linear regime. Is it the case? Have the authors also characterized non-linearity as a function of kinetic inductance?

16- Line 198, "In Fig. 2h, i, the relative frequency shift and the internal quality factors as a function of the applied magnetic fields for resonators with hundreds of photons are plotted". The Internal quality factor is not plotted. Then h is for resonators with different thickness but it is not clear which L_k and (i) is for resonators of different thickness but it is not clear which width and kinetic inductance. Also, the caption does not clarify this point.

17- Line 236-239 when Fig 3a-c is introduced, I think a bit more of description of the device would be very useful. The authors assume somehow that the reader is familiar with double quantum dots, charge qubits, etc but for a broader audience, it would be much better to explain more and describe the device in more detail. In particular, in lines 258-260: "The triple quantum dot array is operated as a DQD formed on the right side, with the two left-most gates grounded." It is very hard to understand to what the authors are referring to. There is no figure to understand which gate is which...

18- Line 251-254, have the authors thought in a way to improve the quality factor of the device?

19- Fig 3d is never called in the text. Is it necessary? Fig 3g is never called in the text.

20- Fig 3e, is the stability diagram compatible with DC measurements? DQD device is not working above $V_L=1100\text{mV}$, $V_R=820\text{mV}$? Or is it the dispersive coupling that is too weak? Why Fig 3f is in a different region?

21- Line 277, "Thus, for a different interdot transition, we probe the system by reflection spectroscopy." Different to what? Why his one and not another one? It seems to me a bit uncomfortable to show one region where there is an anticrossing, give the parameters and not discuss further if for other regions in the stability diagram is similar and on what it depends. Can the authors please comment on this?

22- Line 286, "vacuum rabi splitting" but the resonator is probed with hundreds of photons...

23- Line 289, How is the gamma obtained? From the width?

24- Fig. 4, how come that the signal looks so broad, rather noisy?

25- Line 330, I don't like the wording "ceteris paribus", it doesn't really apply very well.

26- Extended Figure E5, I'm surprised that you don't have isolators to prevent the noise of the HEMT amplifier.

27- Extended Figure E6 very nice and useful.

Version 1:

Reviewer comments:

Reviewer #1

(Remarks to the Author)

Reviewer #2

(Remarks to the Author)

All of my questions and comments have now been convincingly addressed. The revisions made by the authors have improved the overall readability of the manuscript, which I believe is now ready for publication in Nature Communications.

Reply to the Referee report for the manuscript "Strong Charge-Photon Coupling in Planar Germanium Enabled by Granular Aluminium Superinductors" by Janík *et al.*

We thank the referees for their comments and suggestions, shown in blue, to which detailed responses follow below. Changes to the manuscript are highlighted in red in this document. In both the manuscript and the supplementary material, deletions are ~~struck out~~, and additions are marked in red.

1 Reviewer #1

The paper discusses the integration of granular aluminum (grAl) resonators with germanium-based double quantum dots to achieve strong charge-photon coupling. This research is significant for advancing long distance couplings in quantum dot based systems. Granular aluminum, known for its high kinetic inductance, is used to create superconducting coplanar waveguide resonators with high impedance, facilitating strong coupling rates. Key achievements include the development of a high-vacuum-compatible wireless ohmmeter for in situ measurement, allowing precise control over the film's kinetic inductance during deposition. This innovation addresses the challenge of poor reproducibility in high-impedance films. The study demonstrates a strong charge-photon coupling rate of approximately 566 MHz and a cooperativity of 251, using a resonator with a characteristic impedance of $7.9\text{ k}\Omega$. The results pave the way for novel qubit designs and high-fidelity, long-distance two-qubit gates, crucial for scalable quantum computing.

The authors have achieved very nice results addressing a key challenge in using high kinetic inductance films for hybrid systems and shown it can be integrated with SiGe based quantum dots with a strong charge-photon coupling. I believe their results are suitable for publication in *Nature Communications* when concerns below are addressed.

We reply: We sincerely thank the referee for this insightful summary of our paper. We appreciate their recognition of the significance of our work in advancing long-distance couplings in quantum dot-based systems. We are particularly glad that they highlighted the development of the high-vacuum-compatible wireless ohmmeter, as we believe it is a crucial innovation for achieving reproducibility in high-impedance films.

Their acknowledgement of our results, including the charge-photon coupling rate and cooperativity, is encouraging, and we agree that these findings contribute to the potential for novel qubit designs and scalable quantum computing applications. We answer the comments and questions provided in their report and have made the necessary revisions to the manuscript. We thank them for their recommendation for publication in *Nature Communications*.

1.1

How does the kinetic inductance compare to measured R_s , does it follow Mattis-Bardeen formula. If so where does the films sheet inductance of the order $\sim\text{nH/square}$ fall on the SIT of the grAl films. Can you comment on SIT of the grAl films processed using this particular method and process.

We reply: The Mattis-Bardeen formula relates the kinetic inductance L_k with the normal state resistance R_s as $L_k = \hbar R_s / (\pi \Delta)$ [1], where \hbar is the reduced Planck constant and Δ is the superconducting energy gap. The gap Δ can be estimated from the Bardeen-Cooper-Schrieffer (BCS) approximation $\Delta \approx \Delta(T = 0\text{ K}) = 1.76 k_B T_c$ [2], where T_c is the critical temperature.

We have a crude estimate of T_c . We have estimated the critical temperature of the grAl films by measuring the sheet resistance of a grAl strip by a four-probe method while cooling down a dilution refrigerator, as seen in fig. 1. The sheet resistance of the film, shown in fig. 1b, is correlated with the temperature of the mixing chamber, shown in fig. 1a. Similarly to the previous observation reported [3], the critical temperature is weakly decreasing with increasing resistivity from $\approx 2.4\text{ K}$ at $8 \times 10^2\ \mu\Omega\text{ cm}$ to $\approx 1.6\text{ K}$ at $1 \times 10^4\ \mu\Omega\text{ cm}$, as seen in fig. 1c. Considering the error bars, further considerations can be simplified by assuming a critical temperature T_c of 2 K for all samples within the range of interest. A similar critical temperature has been reported for room-temperature evaporated grAl also in ref. [3].

Figure 1: **Evaluation of the critical temperature.** **a** The temperature of the mixing chamber T_{MXC} as measured by the fridge temperature sensor (blue). The T_{MXC} temperature values are translated to the temperature of the sample through the clock time. By assuming that the mixing chamber temperature is the temperature of the sample and by continuously measuring the sheet resistance R_s , the critical temperature T_c can be evaluated as shown in **b**. Since the fridge cooldown is rather abrupt around the critical temperature, the number of resistance points is scarce. The mean and error are taken to lie in the middle between the first zero resistance point and the previous point. **c** Normal state resistivity ρ dependence of the critical temperature T_c . The black crosses represent data points reported in ref. [3]

To evaluate the sheet resistance R_s of the grAl films, we measure a strip of grAl by a four-probe method. It is worth noting that the sheet resistance of the narrow grAl film measured at low temperatures differs from that of a macroscopic test sample measured after evaporation, either by a multimeter or the wireless ohmmeter method. This observation cannot be explained by the change in resistance with decreasing temperature, as the grAl resistivity exhibits only a weak temperature dependence [3]. Rather, it was also observed elsewhere [4] that grAl sheet resistance decreases with decreasing width of the grAl strip. It was hypothesised that due to a narrower channel of resist during the evaporation, the local O_2 pressure decreases, resulting in lower oxidation. This hypothesis would indicate that oxidation happens when aluminium is already at the surface of the sample [4]. As a consequence of using 100 nm wide stripes, most sheet resistances R_s are lower than that of macroscopic test samples with targeted $\sim 2.5 \text{ k}\Omega \square^{-1}$ measured after evaporation by roughly a factor of 2.

Figure 2: **Sheet kinetic inductance dependence on the sheet resistance.** The evaluated sheet kinetic inductance is plotted in blue as a function of the sheet resistance. The red line represents the Mattis-Bardeen formula for kinetic inductance with a critical temperature T_c of 2 K. The shaded area takes into account the critical temperature spread in the range of 1.5–2.5 K. The values are in reasonable agreement.

Putting these results together, the sheet kinetic inductance L_k evaluated from the resonance frequencies, as explained in the main text, can be related to the sheet resistances R_s of the deposited films and compared to the Mattis-Bardeen formula, as seen in fig. 2. The values are in reasonable agreement.

According to the Mattis-Bardeen formula, the BCS gap approximation and a critical temperature T_c of 2 K, sheet inductance of $1 \text{ nH } \square^{-1}$ would require normal state sheet resistance R_s of $\sim 1.6 \text{ k}\Omega \square^{-1}$, which is well below the expected SIT.

We comment on the substrate and process dependence of the SIT in the next reply.

1.2

The authors reference [44] as the measured SIT of their grAl, however, the SIT can be highly substrate-dependent and process dependent: one could imagine different screening effects from a low resistivity substrate. I ask the authors to comment on the SIT specifically of their processes. In particular the reference [44] studied grAl on highly resistive substrates such as sapphire and different surface morphology. How does surface roughness and substrate resistivity impact the grAl properties and its SIT. It seems important to this work and future applications to determine the exact SIT for this work. I recommend additional data or thorough discussion on this.

We reply: We appreciate the referee’s comments on the substrate and process dependence of the superconducting - insulator transition (SIT) in granular aluminum (grAl). While we recognize that the SIT can indeed vary with substrate properties and fabrication processes, we believe that determining the precise SIT is not essential for the specific application presented in this work.

In our study, we fabricated grAl resonators on different substrates, including commercially available Si wafers with resistivities in the range of $1\text{--}5 \Omega \text{ cm}$, and Ge/SiGe heterostructures where the top SiGe spacer, Ge quantum well, and parts of the lower spacer were etched away apart from the mesa region. While the Si substrates have well-defined properties, the exact SIT for these wafers is not essential in the presented context, as they served purely as substrates for the initial characterization of the grAl resonators.

In contrast, the Ge/SiGe heterostructures are less controlled, both in terms of surface roughness, determined by the wafer growth and the specific etching process, and substrate resistivity, which varies based on background doping conditions during heterostructure growth. Therefore, establishing the precise SIT for these substrates would not yield broadly applicable insights, as the transition would vary depending on the specific growth and etching conditions of each heterostructure.

Moreover, the SIT is also highly dependent on the geometry of the resonator, particularly its width [5]. A comprehensive study of the SIT across different substrates and resonator designs is beyond the scope of this paper. Our focus here is to demonstrate precise control over high-impedance resonators and their application in achieving strong charge-photon coupling.

1.3

Do the authors measure the $R_s(T)$ and do they see direct SIT or possibly interrupted by dissipative phases, such as anomalous metal phases. Important for determining origin of dissipation in their system. It seems that sufficient transport characterizations are missing to evaluate the SIT of their grAl.

We reply: We have not conducted systematic measurements of R_s as a function of temperature, but we did not observe any anomalous behaviour indicative of intermediate dissipative phases, such as anomalous metal phases. Instead, our measurements show binary outcomes consistent with either a superconducting, behaving like fig. 1b, or a normal state. We acknowledge that additional transport characterizations would provide more insight into the SIT of our grAl resonators. We appreciate this suggestion for future investigations. However, as explained above, we believe that the determination of SIT for our very specific conditions may have limited relevance to the broader community and, therefore, could be the focus of future work focusing more on the material characterization of grAl rather than its integration with circuit quantum electrodynamics

experiments. Our developed method can nicely facilitate future investigations into this rich area of physics.

1.4

Comment: what is the expected sheet kinetic inductance to possibly be achieved using their grAl process.

We reply: Concerning the maximal sheet kinetic inductance L_k below the SIT, for ~ 20 nm-thin grAl films, the SIT has been measured around $5 \text{ k}\Omega \square^{-1}$ [3]. Assuming the Mattis-Bardeen formula, the BCS gap approximation and a critical temperature T_c of 2 K, the maximal achievable sheet kinetic inductance is $L_k^{\text{max}} = \hbar R_s / (\pi \Delta) \approx \hbar R_s^{\text{max}} / (1.76 \pi k_B T_c) = \hbar (5 \text{ k}\Omega \square^{-1}) / (1.76 \pi k_B (2 \text{ K})) = 3.45 \text{ nH } \square^{-1}$.

1.5

Figure 1 comment: what is the exponential fit, I see no comments or details about the fit. Is the red shaded region above SIT mark inferred from ref [44] or measured?

We reply: The exponential fit shown in Figure 1 is empirical and has been used in previous studies, such as in ref. [6], to model the sheet resistance dependence of granular aluminium (grAl) on the oxygen pressure. The exponential fit yields $R_s [\text{k}\Omega \square^{-1}] = a e^{b \cdot \text{O}_2 [\text{sccm}]}$, where $a = 7.94 \times 10^{-4} \text{ k}\Omega \square^{-1}$ and $b = 1.03 \text{ sccm}^{-1}$. The red-shaded region above the SIT mark is inferred from Ref. [44], as indicated in the figure caption.

1.6

Quality factors are 10^4 , which is ‘high’, however, low compared to other cQED systems and microwave resonator circuits. Can the authors comment on the enhanced loss. Did the authors compare Qs on an insulating substrates.

We reply: While a quality factor of 10^4 is considered sufficient for resonator-quantum dot systems as it would not hinder the resonant charge-photon coupling spectroscopy experiments, it is indeed lower compared to state-of-the-art circuit quantum electrodynamics (cQED) systems and microwave resonator circuits.

The enhanced loss mechanisms in our granular aluminum (grAl) resonators can be attributed to a combination of factors, including intrinsic losses from two-level systems (TLS) and quasiparticles, as well as potential surface roughness and material defects that may arise during fabrication. In fact, as we have reported in previous works [7, 8] with 50Ω Al resonators, Ge itself seems to be a source for losses. This is not fully understood and more work needs to be done in this direction.

Regarding the comparison of quality factors on insulating substrates, we have not conducted a direct comparison in this study. However, we acknowledge that using insulating substrates could influence the loss mechanisms and the overall performance of the resonators. Future work will explore the impact of substrate choice on quality factors to better understand the loss mechanisms in our grAl resonators and potentially identify strategies to enhance their performance.

1.7

Figure 2: in particular the authors see Q change orders of magnitude with power suggesting TLS or quasiparticles. Seems important to comment on intrinsic losses in the resonators and the impact on coherent photon-charge couplings.

We reply: We appreciate the referee’s insightful comment regarding the changes in the quality factor observed with power in Figure 2. These variations indeed suggest the influence of two-level systems (TLS) or quasiparticles, which are known to contribute to intrinsic losses in superconducting resonators. We have included a discussion on these intrinsic losses in the revised manuscript.

The power dependence of the internal quality factor reveals behaviour suggestive of losses due to two-level systems.

1.8

Figure 2 (f): it is very difficult to make out triangles and circles in the plot. Is Si substrate low resistivity substrates?

We reply: Following the reviewer’s suggestion, we have revised the figure and included an enlarged version as an Extended figure to enhance its readability.

Figure 3: **Center conductor width and sheet kinetic inductance dependence of the characteristic impedance.** **a** Center conductor width dependence of the characteristic impedance for different sheet kinetic inductances. **b** Sheet kinetic inductance dependence of the characteristic impedance for different resonator widths.

The resistivity of the silicon substrate is $\rho = 1\text{--}5 \Omega \text{ cm}$. We intentionally chose low resistivity substrates as the grAl resonators are developed for use with semiconductor heterostructures with much higher losses than high resistive silicon or sapphire wafers [7, 9, 10].

1.9

Is the lever arm value 0.2 extracted experimentally or a fit parameter or simulation parameter. This should be clarified bottom p. 5

We reply: Following the referee’s suggestion, we have made the following adjustments to the text.

The coupling strength agrees with the expected value $g_c^e/2\pi = e\beta(\omega_r/2\pi)\sqrt{Z/\pi\hbar}/2 \approx 685$ MHz within less than 20%, with e being the elementary charge, given a differential lever arm $\beta = 0.241$ (Methods), $\omega_r/2\pi = 7.27$ GHz and $Z = 7.9$ k Ω .

To quantify the capacitive effect of the resonator gate on the double quantum dot (DQD) with a differential lever arm, a test sample was fabricated. A DC voltage was connected to the gate mimicking the resonator extension, a DQD was formed, and lever arms were extracted from Coulomb diamonds. The lever arm of the ‘resonator’ gate to the underlying quantum dot was $\alpha_{\text{ResQD}} = 0.265 \pm 0.019$. Since the charge-photon coupling is given by the electrochemical potential difference between the dots induced by the resonator, the α_{ResQD} lever arm needs to be corrected for the effect on the side dot. The cross-lever arm, between the resonator and the side quantum dot, was extracted to be $\alpha_{\text{ResSD}} = 0.024$. The differential lever arm is thus estimated to be $\beta = \alpha_{\text{ResQD}} - \alpha_{\text{ResSD}} = 0.241 \pm 0.019$. The discrepancy between the estimated and measured charge-photon coupling rate could be explained by the lever arms dependence on the number of confined holes, particular lift-off process and DQD tuning.

1.10

The authors claim high internal Q at finite field, but the field data is taken at high photon number ~ 100 . How does the single photon Q change with applied fields. This should be important for applications in QD devices that need large, applied fields to operate.

We reply: We fully agree with the reviewer’s statement that, for many applications, a single photon regime is necessary. The reason we have chosen to operate the resonator in an increased average photon number regime stems from a compromise between the average photon number and the measurement time due to limited experimental resources. The single photon regime usually requires lower measurement bandwidth or averaging due to a decreased signal-to-noise ratio, significantly prolonging the magnetic field sweeps, which could not be afforded. The reason why this does not invalidate our conclusion is the hypothesis that below the threshold field when vortices enter, the magnetic field does not introduce any additional losses. This is supported by the flat dependence of Q_i shown in Extended fig. E4.

Following the referee’s inquiry, we have additionally analysed measurements of internal quality factors with and without applied magnetic field for low and high average intraresonator photon numbers, as shown in fig. 4. We observe, as expected, that until vortices enter, internal quality factors remain unaffected.

Figure 4: **Internal quality factors for low and high average intracavity photon numbers with and without applied magnetic field.** Until vortices enter, the internal quality depends on the average intracavity photon number, but not on the applied magnetic field.

2 Reviewer #2

The manuscript “Strong Charge-Photon Coupling in Planar Germanium Enabled by Granular Aluminium Superinductors” (NCOMMS-24-44187-T), by M. Janík and the rest of G. Katsaro’s group, builds upon results by the same group, who previously investigated (1) the physics of hole qubits in germanium quantum dots (Nat Commun 9, 3902 (2018), Nat. Mater. 20, 1106–1112 (2021)) as well as (2) superconducting circuits implemented on planar germanium substrates (Nat Commun 15, 169 (2024), Nat Commun 15, 6400 (2024)). In the present work, the authors combine these two elements and strongly couple (1) the charge of holes on a Ge double quantum dot (DQD) to (2) a photon on a superconducting resonator on the same substrate. The three main results of this work are:

- While grAl was extensively studied in the past years, this manuscript presents, to my knowledge, the first realization of a hybrid DQD-resonator coupling experiment with grAl. As a result, this device presents the highest impedance reported in such hybrid experiments.
- The authors achieve the highest hole charge-photon coupling strength in the literature. Note, however, that a higher coupling strength was achieved with electrons in Ref. [17] and that a higher cooperativity was reached with holes in Si in Ref. [21]. The current results therefore compete with the state of the art.
- The manuscript introduces a novel method for monitoring the film’s resistivity during deposition and demonstrates that it allows targeting the film’s kinetic inductance reproducibly.

In particular, the reproducibility of the high resonator impedance achieved indicates that this is a solid platform to engineer photon-mediated coupling between distant hole spins. I would moreover like to highlight the clarity and completeness of the introduction, as well as the fantastic work compiling the comprehensive list of related experiments, presented in Tab. E1 and Fig. E6 in the appendix. This summary facilitates direct comparison to similar results in the literature. Moreover, the authors honestly quote the minimum impedance that could be reliably reproduced in multiple devices, rather than highlighting the maximum achieved value, as is often done in the field. For all these reasons, I believe that this article is suitable for publication in Nature Communications. I however recommend adding some clarity to the aspects described below, to improve the manuscript readability.

We reply: We appreciate the thorough review and insightful comments regarding the manuscript. The recognition of the key results, including the first realization of a hybrid DQD-resonator coupling experiment with granular aluminium, the highest hole charge-photon coupling strength in the literature, and the novel method for monitoring film resistivity, is greatly valued.

We are also grateful for the positive remarks on the clarity of the introduction and the comprehensive summary in Tab. E1 and Fig. E6. In response to the recommendations for improving readability, we made the necessary adjustments to enhance clarity.

We thank the referee for the recommendation for publication in *Nature Communications*.

2.1

One of the main claims of the manuscript is the reduction of the standard deviation of R_s when the ohmmeter monitoring is used. The authors focus on discussing the results for Q4 and present Q1, Q2 and Q3 as calibration samples and thus don’t discuss their results quantitatively. What is exactly calibrated in these quadrants? Apart from giving the value of the standard deviation for Q4, it would be great to also state these for Q1, Q2 and Q3. Or is there a reason why their standard deviations are expected to be larger? Similarly, for completeness, it would be great if the thickness distribution for Q1, Q2 and Q3 was also indicated, as it is done for Q4.

We reply: The goal of the calibration procedure, which takes place in the first three quadrants, is to find the optimal oxygen flow, which provides the desired sheet resistance R_s for 25 nm-thick films. To that end, we fix

the deposited thickness in the first three quadrants to 25 nm and gradually adjust the oxygen flow. Due to the nominally fixed thickness in the first three quadrants, the standard deviation is larger. Finally, in the last quadrant, on top of setting the fine-tuned value of oxygen flow, we also release the thickness constraint to achieve improved precision. The statistical analysis of the first three quadrants is presented in table 1.

	#	R_s^{\min}	R_s^{\max}	R_s^{mean} $\text{k}\Omega\text{sq}^{-1}$	R_s^{median}	R_s^{std}	t^{\min}	t^{\max}	t^{mean}	t^{median}	t^{std}
									nm		
w/o	54	0.06	16	2.64	2.08	2.3			~25		
Q1	26	0.25	7.85	1.85	1.31	1.68			~25		
Q2	26	0.99	7.31	3.12	2.65	1.8			~25		
Q3	23	1.12	4.66	2.78	2.7	1.09			~25		
Q4	21	1.68	4.42	2.83	2.61	0.82	18.3	58.7	27.1	25.4	8.8

Table 1: **Summary statistics for grAl depositions.** Number of depositions #, minimum R_s^{\min} , maximum R_s^{\max} , mean R_s^{mean} , median R_s^{median} and standard deviations R_s^{std} of normal state sheet resistance, and minimum t^{\min} , maximum t^{\max} , mean t^{mean} , median t^{median} and standard deviations t^{std} of film thickness for grAl depositions without the wireless ohmmeter holder (w/o) and for the respective quadrants (Q1, Q2, Q3, Q4).

2.2

Related to this point, indicating how many data points were used for each of the distributions in Fig. 1c would be helpful for determining the conclusiveness of the comparison between methods.

We reply: The previous answer addresses this remark.

2.3

Is R_s always zero for O₂ flows below 6 sccm? If it is not, it'd be nice if the y axis in Fig. 1b was plotted in logarithmic scale, to see the difference.

We reply: We appreciate the referee's question regarding R_s for O₂ flows below 6 sccm. R_s is not zero in this range, as can be seen in the figure below with the y-axis plotted on a logarithmic scale. We did not include a logarithmic scale in the manuscript, as a logarithmic scale tends to highlight low values, and the higher values are more relevant for conveying the main message of the figure. Following the suggestion of the referee, we have added the figure as an Extended figure in the revised manuscript.

Figure 5: **Oxygen flow dependence of the sheet resistance.** Oxygen flow dependence of the sheet resistance R_s of a grAl film on a test glass piece with 10 squares as measured with a multimeter after the evaporation.

2.4

How is the grAl resonator coupled to the Nb feedline? How is the coupling capacitance between feedline and resonator determined? It is unclear from the images whether there is a gap or a galvanic connection.

We reply: The grAl resonator is capacitively coupled to the Nb feedline. We evaluate the coupling capacitance between the feedline and the resonator using COMSOL, as stated in the methods section. We have clarified this in the revised manuscript to ensure a better understanding.

Magnitude $|S_{21}|$ of six bare resonators capacitively side-coupled to a common feedline

Figure 2b shows a scanning electron microscope image of a single grAl resonator capacitively side-coupled to a common Nb feedline.

2.5

Fig. 2 presents a great amount of data where, among other quantities, L_k is measured for resonators implemented in films with different R_s . How does the relation between R_s and L_k compare to the theoretical expected dependence?

We reply: We thank the referee for raising this point and we are here providing the same response as given to the first reviewer since both referees raised the same question. Mattis-Bardeen formula relates the kinetic inductance L_k with the normal state resistance R_s as $L_k = \hbar R_s / (\pi \Delta)$ [1], where \hbar is the reduced Planck constant and Δ is the superconducting energy gap. As we do not have a direct experimental probe of the gap Δ , it could be estimated from the Bardeen–Cooper–Schrieffer (BCS) approximation $\Delta \approx \Delta(T = 0\text{ K}) = 1.76 k_B T_c$ [2], where T_c is the critical temperature.

Our experimental setup only allows a crude estimate of T_c . We have estimated the critical temperature of some grAl films by measuring the sheet resistance of a grAl strip by a four-probe method while cooling down a dilution refrigerator, as seen in fig. 6. The sheet resistance of the film, shown in fig. 6b, is correlated with the temperature of the mixing chamber, shown in fig. 6a. Similarly to the previous observation reported [3], the critical temperature is weakly decreasing with increasing resistivity from $\approx 2.4\text{ K}$ at $8 \times 10^2 \mu\Omega\text{ cm}$ to $\approx 1.6\text{ K}$ at $1 \times 10^4 \mu\Omega\text{ cm}$, as seen in fig. 6c. Considering the error bars, further considerations can be simplified by assuming a critical temperature T_c of 2 K for all samples within the range of interest. A similar critical temperature has been reported for room-temperature evaporated grAl also in ref. [3].

Figure 6: **Evaluation of the critical temperature.** **a** The temperature of the mixing chamber T_{MXC} as measured by the fridge temperature sensor (blue). The T_{MXC} temperature values are translated to the temperature of the sample through the clock time. By assuming that the mixing chamber temperature is the temperature of the sample and by continuously measuring the sheet resistance R_s , the critical temperature T_c can be evaluated as shown in **b**. Since the fridge cooldown is rather abrupt around the critical temperature, the number of resistance points is scarce. The mean and error are taken to lie in the middle between the first zero resistance point and the previous point. **c** Normal state resistivity ρ dependence of the critical temperature T_c . The black crosses represent data points reported in ref. [3]

To evaluate the sheet resistance R_s of the grAl films, we measure a strip of grAl by a four-probe method. It is worth noting that the sheet resistance of the narrow grAl film measured at low temperatures differs from that of a macroscopic test sample measured after evaporation, either by a multimeter or the wireless ohmmeter method. This observation cannot be explained by the change in resistance with decreasing temperature, as the grAl resistivity exhibits only a weak temperature dependence [3]. Rather, it was also observed elsewhere [4] that grAl sheet resistance decreases with decreasing width of the grAl strip. It was hypothesised that due to a narrower channel of resist during the evaporation, the local O_2 pressure decreases, resulting in lower oxidation. This hypothesis would indicate that oxidation happens when aluminium is already at the surface of the sample [4]. As a consequence of using 100 nm wide stripes, most sheet resistances R_s are lower than that of macroscopic test samples with targeted $\sim 2.5 \text{ k}\Omega \square^{-1}$ measured after evaporation by roughly a factor of 2, as seen in fig. 7.

Figure 7: **Sheet kinetic inductance dependence on the sheet resistance.** The evaluated sheet kinetic inductance is plotted in blue as a function of the sheet resistance. The red line represents the Mattis-Bardeen formula for kinetic inductance with the critical temperature T_c of 2 K. The shaded area takes into account the critical temperature spread in the range of 1.5–2.5 K. The values are in reasonable agreement.

Putting these results together, sheet kinetic inductance L_k evaluated from the resonance frequencies, as explained in the main text, can be related to the sheet resistances R_s of the deposited films and compared to the Mattis-Bardeen formula, as seen in fig. 7. The values are in reasonable agreement.

2.6

There seem to be error bars on the markers in Fig. 2f, g. Are they extracted from statistics or from error propagation from the measured frequency and width?

We reply: The error bars are extracted from the statistical analysis of several resonators within one sample. We have adjusted the text accordingly to clarify this in the revised manuscript.

The error bars are extracted from a statistical analysis of several resonators within a chip.

2.7

For the inset on Fig. 3a, is it correct that the maroon line represents the potential at the well and that the maroon dot represents a hole delocalized between the two potential wells? It would be helpful if the cartoon was described in the caption.

We reply: The referee is correct - the maroon line represents the potential at the well, and the maroon dot illustrates a hole delocalized between the two potential wells. We have updated the figure caption to provide a clearer description of the cartoon for better understanding.

In this experiment, it was operated as a DQD, as indicated in the schematic cross-section of the device in the inset, which illustrates a hole delocalized in a DQD potential.

2.8

There seem to be voltage offsets between Fig. 3e, Fig. 3f and Fig. E3. Are these due to hysteresis during tuning-up? Otherwise, what's the cause?

We reply: We thank the referee for pointing out the voltage discrepancy between Figs. 3e, f. We had chosen to present a high-resolution scan without switches. Following the comments of the referees, we have realized that it created confusion. Therefore, we have revised the manuscript and replaced the original scans with the corresponding interdot transition scan. Now Figs. 3e, f and Fig. 4 correspond to the same voltage range/interdot transition.

Fig. E3 was taken during a different tuning-up of the device, as it requires a measurable current to flow across the DQD. On the other hand, the rest of the data are taken with the ohmic contacts grounded.

2.9

It would be nice if it was indicated how the dotted and dashed lines in Fig. 4a are obtained. Are they the result of a separate fit? Or are they an overlay of the frequencies resulting from the parameters extracted from Fig. 4b?

We reply: The dotted and dashed lines in Fig. 4a are indeed an overlay of the frequencies resulting from the parameters extracted from Fig. 4b.

The dotted lines show the bare resonator frequency and energies of the bare charge transition, while the dashed lines show the dressed transitions, with frequencies extracted from the line cut fits.

2.10

I find the cooperativity to be a fair quantity for comparing different charge-photon coupling experiments. Including the cooperativity in Tab. E1 or in Fig. E6 would further facilitate direct comparison between experiments.

We reply: We agree that cooperativity is an important parameter for comparing charge-photon coupling experiments. In response, we have now included the cooperativity values in Table E1.

2.11

How are the colors in Fig. E1 determined? They don't seem to always match the underlying crystallinity.

We reply: In Fig. E1, we highlighted in red the areas of the HRTEM image that display Al atomic planes. This figure shows the positions of nanoparticles with atomic planes parallel to the electron beam and perpendicular to the visualization image plane. Consequently, only a portion of the Al crystals are expected to be visible under these conditions. HRTEM is not the most suitable method for displaying the full density of Al nanoparticles, as it only reveals a part of them. However, the EELS composition maps can accurately depict the full density of these nanoparticles embedded in the AlOx surrounding matrix.

We have updated the figure caption to better clarify these points.

HRTEM reveals the positions of nanoparticles with atomic planes parallel to the electron beam and perpendicular to the visualization image plane. Consequently, only a portion of the Al crystals are expected to be visible under these conditions. However, the EELS compositional maps in Fig. 1a accurately depict the full density of these nanoparticles embedded in the AlOx surrounding matrix.

2.12

In lines 159 - 163 it says "In Figs. 2f, g [...] we reproducibly target sheet resistances of 2.5 kOhm/sq". The sheet resistance however varies in Figs. 2f, g, and thus this sentence is confusing.

We reply: We appreciate the referee's observation regarding the statement about the reproducibly targeted sheet resistance. We acknowledge that the sheet inductance, which is related to the sheet resistance, indeed appears to vary in Figs. 2f and 2g. This impression is greatly enhanced by the logarithmic scales. The claim about reproducible resistance targeting refers to the developed wireless ohmmeter holder in the first part of the sentence. Its effect is seen in Fig. 1c. Quantifying the spread of inductance shown in Figs. 2f, g yields a mean of $1.1 \text{ nH } \square^{-1}$ and standard deviation of $0.6 \text{ nH } \square^{-1}$, which is consistent with the spread of resistance considering the uncertainty in the critical temperature. To avoid confusion, we have removed the word "reproducibly" from the referred text.

Leveraging the developed wireless ohmmeter holder, we target sheet resistances of $2.5 \text{ k} \square^{-1}$ and obtain films with a kinetic inductance reaching $(2.7 \pm 0.1) \text{ nH } \square^{-1}$.

2.13

Line 211: what is the difference between the d used here and the w used everywhere else in the manuscript?

We reply: Parameter d in line 211 is the general dimension of the superconductor perpendicular to the applied magnetic field. For a magnetic field applied out-of-plane, this dimension is the width of the center conductor w .

2.14

Line 214: the inset of Fig. 2g is mentioned, but Fig. 2g doesn't seem to have an inset. Should it be Fig. 2h instead?

We reply: Fig. 2g does not contain an inset, so the reference should indeed refer to Fig. 2h, which includes the inset. We have corrected this in the revised manuscript.

2.15

The x tick labels and the y label are not specified in the inset to Fig. 2h.

We reply: We acknowledge that these labels are not specified, which can lead to confusion. We have adjusted the revised manuscript to include clear labelling for both axes in the inset to enhance clarity and understanding.

2.16

Could it be that the variables V_G^R , V_G^L and I_{SD} , and are not explicitly defined. It would be helpful if it was indicated in Fig. a, b or c which gate is which.

We reply: Indeed, these variables were not clearly defined in the manuscript. To enhance clarity, we have now provided labels to Figures 3a, 3b, and 3c to indicate which gate corresponds to each variable.

2.17

Could it be that t_c is not defined in the main text? Same for μ_R , μ_L and κ_c in the appendix.

We reply: We appreciate the referee's observation regarding the definitions of t_c , μ_R , μ_L , and κ_c . These terms were not explicitly defined in the main text and appendix. We have now included their definitions to ensure clarity for the readers.

A clear avoided crossing is observed when the parabolic DQD charge transition frequency equals the frequency of the resonator ($2t_c/h \approx \omega_r/2\pi$, where t_c is the DQD tunnel coupling) ...

... where \hat{a} is the resonator field, \hat{b}_{in} is the input field and κ_c is the resonator coupling rate.

... μ_L (μ_R) is the electrochemical potential of the left (right) quantum dot, and α_{ij} is the lever arm of the gate i on dot j ($i, j \in \{L, R\}$). The plunger ...

2.18

What's the black point in Fig. E6a?

We reply: The black point in Fig. E6a represents a singular occasion of using MoRe (molybdenum-rhenium) disordered superconducting resonator for charge-photon coupling. We have added an item in the figure legend for better clarity.

3 Reviewer #3

Referee report on the manuscript “Strong Charge-Photon Coupling in Planar Germanium Enabled by Granular Aluminium Superinductors” by Janík et al. In this work the authors present a technological development which enables the reproducible and controllable fabrication of high impedance superconducting microwave resonators, in this case using granular aluminum. The authors present a complete characterization of their method and demonstrate a strong coupling between a high impedance resonator and a charge qubit realized in a Ge hole gas. As the authors point-out, high kinetic inductance superconductors are interesting for the realization of qubits, amplifiers, photon detectors, and for the fabrication of high-impedance resonators which may couple strongly to spin and charge qubits, or other mesoscopic systems. Therefore, I totally agree with the authors on the fact that having a method to fabricate reproducible thin films of disordered superconductors with a given sheet resistance is highly desirable. The method presented in this work is very nice and rather simple (a wireless ohmmeter), in this case applied to the evaporation of granular aluminum films, but can be extended to other materials. I find that the characterization of the microwave resonators fabricated with granular Al is rather complete (and useful), and obtaining such large impedances, larger than the resistance quantum, is a clear demonstration of the platform. I think the results give a fair account of the main advantages of the fabricated resonators in term of kinetic inductance, impedance, internal quality factor and magnetic field dependence. The results on the photon-charge coupling are also impressive although the experimental evidence given in the work is scarce, I would have preferred to show different regions in the stability diagram, discuss more in detail the signal, etc. I give in the following a list of comments and questions. I recommend publication in *Nature Communications* after answering my criticism.

We reply: We sincerely thank the referee for the thorough evaluation of our manuscript and the positive feedback regarding our work. We appreciate their recognition of the significance of our technological development and the completeness of our characterization of microwave resonators fabricated with granular aluminium.

We acknowledge their concern regarding the experimental evidence presented in the results on photon-charge coupling. Unfortunately, we could not provide a more comprehensive study as the device exhibited instability during the measurements. This limitation affected our ability to explore different regions in the stability diagram and provide a more detailed discussion of the signal. We have clarified this point in the revised manuscript to ensure transparency regarding the challenges encountered.

We thank the referee for their constructive criticism and recommendation for publication in *Nature Communications*. We have made the necessary revisions in response to their comments.

3.1

In the abstract: “However, fully exploiting the potential of disordered or granular superconductors is challenging, as their inductance and, therefore, impedance at high values are difficult to control.” I find the wording a bit weird. “Impedance at high value hard to control”. I think it is not clear at this point what “control” means. Following: “Here we have integrated a granular aluminium resonator, having a characteristic impedance exceeding the resistance quantum, with a germanium double quantum dot and demonstrate strong charge-photon coupling with a rate of $g_c/2\pi = (566 \pm 2)$ MHz. This was achieved due to the realisation of a wireless ohmmeter, which allows *in situ* measurements during film deposition and, therefore, control of the kinetic inductance of granular aluminium films. Reproducible fabrication of circuits with impedances (inductances) exceeding 13 k Ω (1 nH per square) is now possible.” I think, it would be better in the other order, explain first the problem, the technological solution and then the application. The wording “and therefore,” is used twice and could be avoided by reorganizing the sentences better.

We reply: Following the referee’s suggestion, we rephrased the relevant section to improve readability and ensured the explanation followed a more logical order, presenting the problem first, followed by the solution and its application.

Here, we report a reproducible fabrication of granular aluminium resonators by developing a wireless ohmmeter,

which allows *in situ* measurements during film deposition and, therefore, control of the kinetic inductance of granular aluminium films. Reproducible fabrication of circuits with impedances (inductances) exceeding $13\text{ k}\Omega$ (1 nH per square) is now possible. By integrating a $7.9\text{ k}\Omega$ resonator with a germanium double quantum dot, we demonstrate strong charge-photon coupling with a rate of $g_c/2\pi = (566 \pm 2)\text{ MHz}$.

3.2

Lines 29 and 31, achieving and achieved are used one after the other. Lines 27-29 long-distance two-qubit gates is repeated and this could be avoided.

We reply: For better clarity, we have replaced the repeating terms.

Currently, one of the main challenges for improving the fidelity of such gates lies in increasing the spin-photon coupling strength [11].

3.3

Lines 39-40 “Outside the field of hybrid devices” it is not clear at this point what does hybrid refers to.

We reply: In this context, "hybrid" refers to devices that combine superconductors and semiconductors. The text is revised to include this definition for clarity.

Outside the field of quantum dot cQED devices, much higher impedance has been achieved...

3.4

Line 39, it is written “extended Fig“. This repeat in many places in the text.

We reply: We thank the referee for bringing this to our attention. We have corrected it throughout the manuscript.

3.5

Line 81, on Granular Al, I love Fig 1a and extended fig E1. In particular this later I find it incredible.

We reply: We agree with the reviewer that the TEM analysis is important for understanding the nature of GrAl.

3.6

Line 110. What does “cQED samples” mean?

We reply: For better clarity, we have replaced "cQED samples" with "quantum dot-resonator samples" in the revised manuscript to ensure that all readers can understand the terminology.

... which shows individual evaporations of bare resonator and quantum dot-resonator samples targeting $2.5\text{ k}\Omega\text{ }\square^{-1}$.

3.7

Line 121, “ Such a procedure yields a reproducible resistance at the expense of the uncertainty in the final film thickness.” As I understand this makes sense, because the feedback is on the resistance. Tuning with three test

samples can reduce the error, but there is no control on the thickness. The sentence in the text is “ As shown in Fig. 1c, this procedure significantly reduces the uncertainty of both resistance and thickness, making it possible to target the desired resonance frequency range.” But in the figure there is a large dispersion in the thickness, which is also mentioned in the caption: “With the holder, the resulting sheet resistance dispersion is significantly decreased at the expense of the defined film thickness, as shown in the inset.” All in all, there is some inconsistency. In addition, Fig. 1c is not described in the main text, where the dispersion among the quadrants come from?

We reply: We thank the referee for highlighting the confusion. To resolve the inconsistency, we have removed the word “thickness” from the sentence "As shown in ... " to focus on the reproducibility of the resistance, which is the main result of the procedure. The dispersion in film thickness does remain significant, as correctly pointed out, and we have updated the text accordingly.

The dispersion among the quadrants, even for nominally identical consecutive evaporations, is presumably attributed to the unstable evaporation rate caused by the oxygen contamination of the aluminium source according to ref. [6]. We have not clearly determined the source of the dispersion.

Fig. 1c is referenced twice in the main text, and its extensive description is provided in the figure caption, therefore we prefer not to describe it again in the main text.

3.8

I find Fig.2 a bit hard to read. There are three different color scales which are used for different things (Lk, w, thickness) in some cases with the same colors. I would suggest a strong revision of this figure to make it more friendly with the reader. Maybe separating with subtitles, or using disjoint color scales...

We reply: We recognize that the use of multiple color scales can make the figure difficult to interpret. To improve readability, we have replotted the figure as an Extended figure with an increased size. This addition aims to enhance the overall clarity and user-friendliness of the figure.

Figure 8: **Center conductor width and sheet kinetic inductance dependence of the characteristic impedance.** **a** Center conductor width dependence of the characteristic impedance for different sheet kinetic inductances. **b** Sheet kinetic inductance dependence of the characteristic impedance for different resonator widths.

3.9

I would appreciate a more precise description of the difference between the devices. In the text you only mention: “Two strategies are employed in order to maximise the impedance: first, increasing the kinetic inductance via the sheet resistance and, second, decreasing the width of the center conductor from 10 μm to 100 nm.” There are six resonators per chip. I guess all of them have the same sheet resistance and thickness. Do they have different central conductor width? Or different length? What does make the frequency difference? How large is the dispersion of the resulting L_k comparing the resonators of the same chip? What about the quality factor dispersion within a chip?

We reply: All resonators on a chip have the same central conductor width and sheet resistance, with the frequency difference arising from variations in the resonator length. The dispersion in kinetic inductance (L_k) between resonators on the same chip is minimal, as seen in the small error bars in Fig. 2f, g, which come from statistical analysis of resonators within a chip. In the same manner, the dispersion of the internal quality factor within a chip is considered in the error bars of Fig. 2e. The internal quality factors of individual resonators used to construct Fig. 2e are plotted below.

Figure 9: **Quality factors of individual resonators used for creating fig. 2e.** Data from 200 nm- and 100 nm-wide resonators are shown in **a** and **b**, respectively.

3.10

In the description of the resonators, we don't see the coupling to the transmission line. How big it is?

We reply: We thank the referee for raising this point and are providing the same response as given to the first reviewer since both referees raised the same question. The grAl resonator is capacitively coupled to the Nb feedline. We evaluate the coupling capacitance between the feedline and the resonator using COMSOL, as stated in the methods section. We have clarified this in the revised manuscript to ensure a better understanding. The magnitude of coupling capacitance is in the order of units of fF.

Magnitude $|S_{21}|$ of six bare resonators capacitively side-coupled to a common feedline

Figure 2b shows a scanning electron microscope image of a single grAl resonator capacitively side-coupled to a common Nb feedline.

3.11

Fig. 2d is never called in the text.

We reply: We have now inserted a reference to Figure 2d in the text at the appropriate location.

The hanger geometry (Fig. 2a, b) is preferred since it allows testing multiple resonators per chip, as seen in Fig. 2d, as well as the precise determination of internal losses and coupling rates.

3.12

In the caption of Fig 2 : “Optical and b, c scanning electron microscope images of six 200nm-wide grAl resonators side-coupled to a feedline”, b and c correspond to a single resonator.

We reply: We have revised the caption to clarify that b and c correspond to a single resonator, and we apologize for the oversight.

a Optical microscope image of six 200 nm-wide grAl resonators side-coupled to a feedline. Due to large sheet kinetic inductance L_k , their length does not exceed 200 μm . The Nb ground plane around the grAl resonator is patterned with a vortex-trapping hexagonal array of circular holes with ~ 150 nm diameter [12]. **b, c** Scanning electron microscope image of a single resonator.

3.13

Line 159, for the description of Figs. 2f, g, It is not clear in the main text how the value of the L_k is obtained. The explanation is given in the methods, but since Eq 1 is there and since the following discussion is on L_k , it would be much better to explain how this value is extracted from the measurement of the resonances. Additionally, have the authors tried to fit the participation ratio from the temperature dependence of the resonance frequency?

We reply: We appreciate the referee’s suggestion for clarity. We had initially expanded the description in the main text to explain how the value of the kinetic inductance (L_k) is obtained from the resonance measurements, but this disrupted the flow of the text. Therefore, we have moved the detailed discussion to the Methods section. Regarding the participation ratio, we did not perform any temperature-dependence experiments.

3.14

Line 170, “However, we observe a limitation with three 100 nm-wide samples, with most devices failing to respond to microwave excitation.” Why is this? I don’t understand. And then... “Nevertheless, other 100 nm-wide samples demonstrate reliable performance”. This is all vague.

We reply: The reason some of the 100 nm-wide samples did not respond to microwave excitation remains unclear. Despite similar fabrication processes, there appears to be variability in the device RF performance for 100 nm-wide samples, and also in switching currents measured by a DC four-probe method. We currently speculate that for such small widths, the background pressure of the evaporator could play a significant role, and sometimes the 100 nm-wide lines are electrically not continuous. We have revised the text to acknowledge this uncertainty and improve the clarity of the explanation.

While the reason remains unclear, we speculate that for the smallest widths, the evaporator background pressure could play a role.

3.15

Line 182, when discussion quality factor. Have you observed that quality factor decreases with kinetic inductance (sheet resistance)? Have you observed that quality factor is affected by film thickness? I guess the gray points in Fig 2e correspond to the non-linear regime. Is it the case? Have the authors also characterized non-linearity as a function of kinetic inductance?

We reply: We have not observed clear dependences of the quality factor on kinetic inductance (sheet resistance) or film thickness.

The grey points in Fig. 2e correspond to the non-linear regime, but the evaluated self-Kerr coefficient K_{11} does not directly depend on the kinetic inductance. Rather, as derived in ref. [13], the self-Kerr non-linearity K_{11} is given as

$$K_{11} = \frac{3}{16} \pi e a \frac{\omega_r^2}{j_c(\rho) V_{\text{grAl}}}, \quad (1)$$

where e is the elementary charge, a is the grain size, ω_r is the resonance frequency, $j_c(\rho)$ is the critical current, which is a function of resistivity ρ , and V_{grAl} is the grAl volume.

3.16

Line 198, “In Fig. 2h, i, the relative frequency shift and the internal quality factors as a function of the applied magnetic fields for resonators with hundreds of photons are plotted”. The Internal quality factor is not plotted. Then h is for resonators with different thickness but it is not clear which L_k and (i) is for

resonators of different thickness but it is not clear which width and kinetic inductance. Also, the caption does not clarify this point.

We reply: The internal quality factor dependence is indeed not provided in the main text, but in the supplementary material as the Extended Figure E4.

In Fig. 2h, i (Extended Fig. 4), the relative frequency shift (the internal quality factors) as a function of the applied magnetic fields for resonators with hundreds of photons is plotted.

Additionally, we have clarified that the kinetic inductance L_k does not influence the resonator's response to the applied magnetic field.

The kinetic inductance L_k does not influence the resonator's response to the applied magnetic field.

3.17

Line 236-239 when Fig 3a-c is introduced, I think a bit more of description of the device would be very useful. The authors assume somehow that the reader is familiar with double quantum dots, charge qubits, etc but for a broader audience, it would be much better to explain more and describe the device in more detail. In particular, in lines 258-260: "The triple quantum dot array is operated as a DQD formed on the right side, with the two left-most gates grounded." It is very hard to understand to what the authors are referring to. There is no figure to understand which gate is which...

We reply: We recognize that a more thorough explanation would benefit readers who may not be familiar with the mentioned concepts. In the revised manuscript, we have added a more detailed description in the corresponding figure to visually illustrate the gate layout and clarify the operation of the triple quantum dot array, specifically the formation of the DQD on the right side with the left-most gates grounded.

3.18

Line 251-254, have the authors thought in a way to improve the quality factor of the device?

We reply: As previous works of the group have identified Ge as a significant source of losses [7, 8], using flip-chip will be a way to improve the device quality factor [14].

3.19

Fig 3d is never called in the text. Is it necessary? Fig 3g is never called in the text.

We reply: We thank the referee for pointing this out. We have inserted references to Fig. 3d and Fig. 3g in the text at the appropriate locations. Fig. 3d serves as informative support, and while it is not essential, we believe it provides a helpful basis for comparison between the bare resonators and quantum dot-resonator samples.

The quality factors are significantly smaller compared to bare resonators, as seen in Fig. 3d.

Line cuts along the right gate voltage, shown in Fig. 3g, translated...

3.20

Fig 3e, is the stability diagram compatible with DC measurements? DQD device is not working above $V_L=1100\text{mV}$, $V_R=820\text{mV}$? Or is it the dispersive coupling that is too weak? Why Fig. 3f is in a different region?

We reply: As the ohmic contacts were grounded after the initial DC characterization, we do not probe the DC response of the device and, therefore, cannot conclude anything about the DC compatibility of DC and RF measurements in the studied range. For example, Fig. E3 was taken during a different tuning-up of the device, as it requires a measurable current to flow across the DQD, while the rest of the data are taken with the ohmic contacts grounded.

For the particular voltage configuration shown in Fig. 3e we did not observe any further resonator response beyond $V_G^L = 1100\text{mV}$ and $V_G^R = 820\text{mV}$. However, this does not constitute an experimental proof of fully depleting the DQD from charges.

Concerning Fig. 3f, we had chosen to present a high-resolution scan without switches. Following the comments of the referees, we have realized that it created confusion. Therefore, we have revised the manuscript and replaced the original scans with the corresponding interdot transition scan. Now Figs. 3e, f and Fig. 4 correspond to the same voltage range/interdot transition.

3.21

Line 277, “Thus, for a different interdot transition, we probe the system by reflection spectroscopy.” Different to what? Why his one and not another one? It seems to me a bit uncomfortable to show one region where there is an anticrossing, give the parameters and not discuss further if for other regions in the stability diagram is similar and on what it depends. Can the authors please comment on this?

We reply: The previous answer addresses part of this remark remark.

Concerning the analysis of different interdot transitions, we could not conduct detailed investigations due to the instability of the device. In the figure below, we show the scans of four neighbouring interdot transitions for a given interdot tunnel coupling t_c . We observe a charge-photon interaction for all of them. However, since the charge transition frequency is far from being resonant with the resonator, we cannot reliably evaluate the charge-photon coupling strength. By trying to bring the charge transitions shown in **b**, **c**, and **e** resonant with the resonator by tuning the interdot tunnel coupling t_c with the middle tunnel barrier voltage, the device becomes very unstable. This instability prevented us from conducting a systematic investigation of the remaining charge transitions. We have clarified this limitation in the revised manuscript and discussed the impact of device instability on the observed phenomena.

Figure 10: **Interdot and charge-photon interaction spectroscopy scans for neighbouring charge transitions.** **a** Interdot scans for four neighbouring charge transitions within the range shown in Fig. 3e. The blue scan is also shown in Fig. 3f. **b**, **c**, **d**, **e** Corresponding spectroscopy scans across the DQD detunings for a given value of the middle tunnel barrier voltage. We observe a charge-photon interaction for all of them. In **b** and **d** (**c** and **e**), the charge transition frequency is below (above) the resonance frequency. However, since the charge transition frequency is far from being resonant with the resonator, we cannot reliably evaluate the charge-photon coupling strength. By trying to bring charge transitions shown in **b**, **c**, and **e** resonant with the resonator by tuning the interdot tunnel coupling t_c with the middle tunnel barrier voltage, the device becomes very unstable. This instability prevented us from conducting a systematic investigation of the remaining charge transitions. The arrow in **e** indicates an electrical switch.

3.22

Line 286, “vacuum rabi splitting” but the resonator is probed with hundreds of photons. . .

We reply: We agree with the referee that if the resonator were probed with hundreds of photons, vacuum Rabi splitting would not occur. However, the resonator was probed well below one intracavity photon,

which can be seen from the following considerations. The average intracavity photon number for a single-port resonator at resonance is calculated as

$$\langle n \rangle = \frac{4}{\hbar \omega_r} \frac{\kappa_c}{\kappa^2} P_d, \quad (2)$$

where \hbar is the reduced Planck constant, ω_r is the resonance frequency, κ (κ_c) is the resonator field decay (coupling) rate, and P_d is the drive power after accounting for input attenuation. The output power of the VNA was set to be -2 dBm, while the nominal line attenuation was 120 dB, as seen in the measurement setup scheme (Fig. E5). After considering the extra attenuation coming from the cables and connectors (of the order of 20 dB), the estimated average intracavity photon number is $\sim 0.2 < 1$.

3.23

Line 289, How is the gamma obtained? From the width?

We reply: The charge decoherence rate $\gamma/2\pi$ of (297 ± 9) MHz was obtained from fitting the zero detuning linecut (red) in Fig. 4b with Eq. 12.

3.24

Fig. 4, how come that the signal looks so broad, rather noisy?

We reply: With the charge transition resonant with the resonator ($2t_c/h = \omega_r/2\pi$), the width of the peaks is determined by the equally weighted photon and charge decay rates, which contributes to the observed broadness of the signal, as the charge decoherence γ is rather large ($\gamma = (297 \pm 9)$ MHz). Additionally, the observed noise is attributed to the instability of our sample, which can affect the clarity of the signal.

3.25

Line 330, I don't like the wording "ceteris paribus", it doesn't really apply very well.

We reply: We agree that the phrase "ceteris paribus" may not be the most appropriate choice in this context. We have revised the wording to improve clarity and better convey the intended meaning.

Using a grAl resonator, given the state-of-the-art quantum dot technologies, spin-photon coupling could enter the ultra-strong coupling regime, following in the footsteps of the superconducting qubits and opening an avenue towards unexplored physics [15] and advanced quantum information processing applications.

3.26

Extended Figure E5, I'm surprised that you don't have isolators to prevent the noise of the HEMT amplifier.

We reply: We appreciate the referee's observation regarding the absence of isolators to prevent noise from the HEMT amplifier in Extended Figure E5. We would like to clarify that there is a circulator present between the sample and the HEMT amplifier to prevent noise. We apologize for the oversight, as the circulator was inadvertently omitted from the schematic. We have now corrected this in the revised Extended Figure E5.

3.27

Extended Figure E6 very nice and useful.

We reply: We thank the referee for their appreciation of our comprehensive overview of the charge-photon coupling literature. We are glad that Extended Figure E6 was found useful for the community, as intended.

-
- [1] Mattis, D. C. & Bardeen, J. Theory of the anomalous skin effect in normal and superconducting metals. *Physical Review* **111**, 412 (1958).
 - [2] Bardeen, J., Cooper, L. N. & Schrieffer, J. R. Theory of superconductivity. *Physical review* **108**, 1175 (1957).
 - [3] Levy-Bertrand, F. *et al.* Electrodynamics of granular aluminum from superconductor to insulator: Observation of collective superconducting modes. *Physical Review B* **99**, 094506 (2019).
 - [4] Fechant, M. *Non-linear lattices of granular aluminium resonators*. Ph.D. thesis, universit  Paris-Saclay (2021).

- [5] Voss, J. N. *et al.* Eliminating quantum phase slips in superconducting nanowires. *ACS nano* **15**, 4108–4114 (2021).
- [6] Rotzinger, H. *et al.* Aluminium-oxide wires for superconducting high kinetic inductance circuits. *Superconductor Science and Technology* **30**, 025002 (2016).
- [7] Valentini, M. *et al.* Parity-conserving Cooper-pair transport and ideal superconducting diode in planar germanium. *Nature Communications* **15**, 169 (2024).
- [8] Sagi, O. *et al.* A gate tunable transmon qubit in planar Ge. *Nature Communications* **15**, 6400 (2024).
- [9] Mi, X. *et al.* Circuit quantum electrodynamics architecture for gate-defined quantum dots in silicon. *Applied Physics Letters* **110** (2017).
- [10] Harvey-Collard, P. *et al.* On-chip microwave filters for high-impedance resonators with gate-defined quantum dots. *Physical Review Applied* **14**, 034025 (2020).
- [11] Dijkema, J. *et al.* Two-qubit logic between distant spins in silicon. *arXiv preprint arXiv:2310.16805* (2023).
- [12] Kroll, J. G. *et al.* Magnetic-field-resilient superconducting coplanar-waveguide resonators for hybrid circuit quantum electrodynamics experiments. *Physical Review Applied* **11**, 064053 (2019).
- [13] Maleeva, N. *et al.* Circuit quantum electrodynamics of granular aluminum resonators. *Nature communications* **9**, 3889 (2018).
- [14] Hinderling, M. *et al.* Direct microwave spectroscopy of Andreev bound states in planar Ge Josephson junctions. *PRX Quantum* **5**, 030357 (2024).
- [15] Forn-Díaz, P., Lamata, L., Rico, E., Kono, J. & Solano, E. Ultrastrong coupling regimes of light-matter interaction. *Reviews of Modern Physics* **91**, 025005 (2019).

The manuscript “Strong Charge-Photon Coupling in Planar Germanium Enabled by Granular Aluminium Superinductors” (NCOMMS-24-44187-T), by M. Janík and the rest of G. Katsaro’s group, builds upon results by the same group, who previously investigated (1) the physics of hole qubits in germanium quantum dots (Nat Commun 9, 3902 (2018), Nat. Mater. 20, 1106–1112 (2021)) as well as (2) superconducting circuits implemented on planar germanium substrates (Nat Commun 15, 169 (2024), Nat Commun 15, 6400 (2024)). In the present work, the authors combine these two elements and strongly couple (1) the charge of holes on a Ge double quantum dot (DQD) to (2) a photon on a superconducting resonator on the same substrate.

The three main results of this work are:

- While grAl was extensively studied in the past years, this manuscript presents, to my knowledge, the **first realization** of a hybrid DQD-resonator coupling experiment **with grAl**. As a result, this device presents the **highest impedance** reported in such hybrid experiments.
- The authors achieve the **highest hole charge-photon coupling strength** in the literature. Note, however, that a higher coupling strength was achieved with electrons in Ref. [17] and that a higher cooperativity was reached with holes in Si in Ref. [21]. The current results therefore compete with the state of the art.
- The manuscript introduces a novel method for monitoring the film’s resistivity during deposition and demonstrates that it allows targeting the film’s kinetic inductance **reproducibly**.

In particular, the reproducibility of the high resonator impedance achieved indicates that this is a solid platform to engineer photon-mediated coupling between distant hole spins. I would moreover like to highlight the clarity and completeness of the introduction, as well as the fantastic work compiling the comprehensive list of related experiments, presented in Tab. E1 and Fig. E6 in the appendix. This summary facilitates direct comparison to similar results in the literature. Moreover, the authors honestly quote the minimum impedance that could be reliably reproduced in multiple devices, rather than highlighting the maximum achieved value, as is often done in the field. For all these reasons, I believe that this article is suitable for publication in Nature Communications. I however recommend adding some clarity to the aspects described below, to improve the manuscript readability:

Questions:

- One of the main claims of the manuscript is the reduction of the standard deviation of R_s when the ohmmeter monitoring is used. The authors focus on discussing the results for Q4 and present Q1, Q2 and Q3 as calibration samples and thus don’t discuss their results quantitatively. What is exactly calibrated in these quadrants? Apart from giving the value of the standard deviation for Q4, it would be great to also state these for Q1, Q2 and Q3. Or is there a reason why their standard deviations are expected to be larger? Similarly, for completeness, it would be great if the thickness distribution for Q1, Q2 and Q3 was also indicated, as it is done for Q4.
- Related to this point, indicating how many data points were used for each of the distributions in Fig. 1c would be helpful for determining the conclusiveness of the comparison between methods.

- Is R_s always zero for O2 flows below 6 sccm? If it is not, it'd be nice if the y axis in Fig. 1b was plotted in logarithmic scale, to see the difference.
- How is the grAl resonator coupled to the Nb feedline? How is the coupling capacitance between feedline and resonator determined? It is unclear from the images whether there is a gap or a galvanic connection.
- Fig. 2 presents a great amount of data where, among other quantities, L_k is measured for resonators implemented in films with different R_s . How does the relation between R_s and L_k compare to the theoretical expected dependence?
- There seem to be error bars on the markers in Fig. 2f, g. Are they extracted from statistics or from error propagation from the measured frequency and width?
- For the inset on Fig. 3a, is it correct that the maroon line represents the potential at the well and that the maroon dot represents a hole delocalized between the two potential wells? It would be helpful if the cartoon was described in the caption.
- There seem to be voltage offsets between Fig. 3e, Fig. 3f and Fig. E3. Are these due to hysteresis during tuning-up? Otherwise, what's the cause?
- It would be nice if it was indicated how the dotted and dashed lines in Fig. 4a are obtained. Are they the result of a separate fit? Or are they an overlay of the frequencies resulting from the parameters extracted from Fig. 4b?
- I find the cooperativity to be a fair quantity for comparing different charge-photon coupling experiments. Including the cooperativity in Tab. E1 or in Fig. E6 would further facilitate direct comparison between experiments.
- How are the colors in Fig. E1 determined? They don't seem to always match the underlying crystallinity.

Minor comments:

- In lines 159 - 163 it says "In Figs. 2f, g [...] we reproducibly target sheet resistances of 2.5 kOhm/sq". The sheet resistance however varies in Figs. 2f, g, and thus this sentence is confusing.
- Line 211: what is the difference between the d used here and the w used everywhere else in the manuscript?
- Line 214: the inset of Fig. 2g is mentioned, but Fig. 2g doesn't seem to have an inset. Should it be Fig. 2h instead?
- The x tick labels and the y label are not specified in the inset to Fig. 2h.
- Could it be that the variables $V_{G'}^R$, V_G^R and I_{SD} are not explicitly defined. It would be helpful if it was indicated in Fig. a, b or c which gate is which.
- Could it be that t_c is not defined in the main text? Same for μ_R , μ_L and κ_c in the appendix.
- What's the black point in Fig. E6a?